# Oestrogen-dependent hypothalamic oxytocin expression with changes in feeding and body weight in female rats

Kazuaki Nishimura [1,2], Kiyoshi Yoshino[2], Naofumi Ikeda[1], Kazuhiko Baba[1], Kenya Sanada[1], Yasuki Akiyama[1], Haruki Nishimura[1], Kentaro Tanaka[1], Satomi Sonoda[1], Hiromichi Ueno[1], Mitsuhiro Yoshimura[1], Takashi Maruyama[1], Toru Hachisuga[3] & Yoichi Ueta [1✉]

Oxytocin (OXT) is produced in the hypothalamic nuclei and secreted into systemic circulation from the posterior pituitary gland. In the central nervous system, OXT regulates behaviours including maternal and feeding behaviours. Our aim is to evaluate whether oestrogen regulates hypothalamic OXT dynamics. Herein, we provide the first evidence that OXT dynamics in the hypothalamus vary with sex and that oestrogen may modulate dynamic changes in OXT levels, using OXT-mRFP1 transgenic rats. The fluorescence intensity of OXT-mRFP1 and expression of the *OXT* and *mRFP1* genes in the hypothalamic nuclei is highest during the oestrus stage in female rats and decreased significantly in ovariectomised rats. Oestrogen replacement caused significant increases in fluorescence intensity and gene expression in a dose-related manner. This is also demonstrated in the rats' feeding behaviour and hypothalamic Fos neurons using cholecystokinin-8 and immunohistochemistry. Hypothalamic OXT expression is oestrogen-dependent and can be enhanced centrally by the administration of oestrogen.

[1] Department of Physiology, School of Medicine, University of Occupational and Environmental Health, Kitakyushu 807-8555, Japan. [2] Department of Obstetrics and Gynecology, School of Medicine, University of Occupational and Environmental Health, Kitakyushu 807-8555, Japan. [3] Department of Obstetrics and Gynecology, Steel Memorial Yawata Hospital, Kitakyushu 805-8508, Japan. ✉email: yoichi@med.uoeh-u.ac.jp

Oxytocin (OXT) is produced in the paraventricular (PVN) and supraoptic nuclei (SON) of the hypothalamus[1]. Peripheral OXT is a neurohypophysial hormone that is originally synthesised in the magnocellular PVN (mPVN) and SON, and is secreted from the posterior pituitary gland (PP) into the systemic circulation. In the periphery, OXT regulates parturition and lactation[2]. Recent studies have suggested that in addition to its peripheral effects, hypothalamic OXT produced in the mPVN and SON acts within the central nervous system to regulate many functions[3], including social recognition and trust-building[4,5]. In addition, OXT is produced in the dorsal parvocellular PVN (dpPVN) and in this context, is involved in the modulation of stress and pain[6,7]. Interestingly, the OXT pathway from the PVN is involved in the control of feeding[8]. In particular, hypothalamic OXT has an anorectic action; therefore, modulating this pathway is anticipated to reduce obesity and high blood glucose levels[9,10]. However, while the peripheral actions of OXT in pregnant and lactating females are well known[11], the sex differences in hypothalamic OXT dynamics remain unclear.

Oestrogen is produced in the ovaries and placenta, and binds to systemic oestrogen receptors (ER) via blood to produce oestrogenic activity[12]. Oestrogen plays an important role in maintaining the physiological functions of systemic organs. Additionally, through the oestrus cycle, oestrogen further regulates female reproductive functions. In particular, oestrogen replacement therapy during menopause has been demonstrated to prevent various diseases as well as treat menopausal disorders in women[13–15]. Furthermore, the effects of oestrogen on food intake are thought to be mediated through ERs (ERα and ERβ) within the central nervous system[16–18]. ERβs are located on OXT neurons and are the predominant ER subtype in the PVN, a hypothalamic area involved in eating[19]. However, details regarding the correlation between oestrogen and hypothalamic OXT dynamics are unknown.

In the present study, we first assessed the effects of sex and oestrogen on body weight and body fat mass in female rats using micro-computed tomography (micro-CT)[20]. We used reporter OXT-monomeric red fluorescent protein 1 (mRFP1) transgenic rats to visualise OXT expression and clarify the relationship between hypothalamic OXT and the oestrus cycle[21,22]. To confirm the relationship between oestrogen and OXT, we assessed the expression of both OXT-mRFP1 and the *OXT-mRFP1* gene in bilaterally ovariectomised (OVX) rats with or without exogenous oestrogen replacement. Then, we assessed whether oestrogen replacement and OVX impact the regulation of OXT produced by hypothalamic neurons, thereby regulating central nervous system functions. We evaluated differences in the expression of the hypothalamic *OXT* and *mRFP1* genes in OVX rats with or without exogenous oestrogen replacement using in situ hybridisation histochemistry. We also assessed food consumption and hypothalamic OXT Fos-neuronal activity in OVX rats with or without oestrogen replacement and with or without intraperitoneal (i.p.) administration of cholecystokinin (CCK)-8, an agent known to selectively activate OXT neurons[23]. Further, we assessed food consumption in OVX and oestrogen replacement rats with i.p. administration of CCK-8 and intracerebroventricular (i.c.v.) administration of OXT receptor antagonist (OXTR-A)[24–26]. Thus, we aimed to investigate whether oestrogen could regulate and control hypothalamic OXT dynamics.

## Results

**Relationship between body weight, fat mass, and feeding.** We observed a significant change in body weight depending on sex, OVX, and oestrogen replacement. Wistar rats were divided into five groups: sham-operated male, sham-operated female, only OVX, OVX + low β-oestradiol (E2) replacement, and OVX + high E2 replacement groups (Exp. A). We observed a significant change in body weight depending on sex and OVX. Female rats with OVX displayed a significant change in body weight compared to sham-operated reproductive female rats, and oestrogen supplementation affected body weight in a dose-dependent manner (Fig. 1a). The difference in body weight was determined by assessing the amount of visceral and subcutaneous fat using micro-CT (Fig. 1b, e, h). There was a difference in fat mass between the 9th (Fig. 1b, c), 14th (Fig. 1e, f), and 16th (Fig. 1h, i) week. We observed a significant change in feeding depending on sex, OVX, and oestrogen replacement (Fig. 1d, g, j).

**OXT-mRFP1 fluorescence differences between female and male rats.** We used adult male and female OXT-mRFP1 Wistar transgenic rats that were maintained under normal laboratory conditions (12-h light, 12-h dark cycle) with free access to food and drinking water[21,22]. We first aimed to ascertain differences in OXT-mRFP1 fluorescence between 10-week-old female rats undergoing a normal oestrus cycle and 10-week-old male rats (Exp. B). Female OXT-mRFP1 transgenic rats were further divided based on the four oestrus stages (pro-oestrus, oestrus, metoestrus, and dioestrus stages). Similar to a previous study, we observed the entire hypothalamus [SON, anterior parvocellular PVN (apPVN), dpPVN, and mPVN] and PP in OXT-mRFP1 transgenic rats using high-power fluorescence microscopy[27] (Fig. 2a). OXT-mRFP1 fluorescence in the SON, apPVN, dpPVN, and mPVN was significantly different between reproductive males and females in the oestrus stage. OXT-mRFP1 fluorescence in the apPVN, dpPVN, and mPVN was significantly different among females depending on the oestrus stage. There was a significant difference in the OXT-mRFP1 fluorescence in the PP between male rats and female rats in the oestrus stage (Fig. 2b). These results suggest that the presence of OXT expression in the hypothalamus and pituitary gland is influenced by sex.

**Effects of OVX on OXT-mRFP1 fluorescence.** OVX and sham operations were performed on the 11th week, and experiments were conducted on the 15th week. To further investigate the influence of sex on OXT expression, OVX (Exp. C) was performed to induce an oestrogen-deficient state in reproductive female OXT-mRFP1 transgenic rats. These rats were compared to a sham-operated control group, which consisted of both male and female rats. As in Exp. B, OXT-mRFP1 fluorescence in the SON, apPVN, dpPVN, and mPVN revealed a significant difference between the male rats and female rats in the oestrus stage (Fig. 3a). Moreover, OXT-mRFP1 fluorescence in the hypothalamus (SON, apPVN, dpPVN, and mPVN) and PP were significantly decreased in OVX rats when compared to females in the oestrus stage, and were similar to the levels in male rats (Fig. 3b). Based on these results, we determined that OVX resulted in a decrease in the OXT expression in the hypothalamus and pituitary gland.

**Effects of oestrogen replacement on OXT-mRFP1 fluorescence.** We performed OVX in 11-week-old female OXT-mRFP1 transgenic rats, conducted hormone replacement on week 15, and performed experiments on week 16. Considering that oestrogen levels were expected to be affected by OVX, oestrogen supplementation experiments were performed (Exp. D). Oestrogen was supplemented in female OVX OXT-mRFP1 rats. Among the OVX groups, the rats in the groups with supplementation of low E2 and high E2 elicited significant changes in OXT-mRFP1 levels in the hypothalamus (SON, apPVN, dpPVN, and mPVN) and

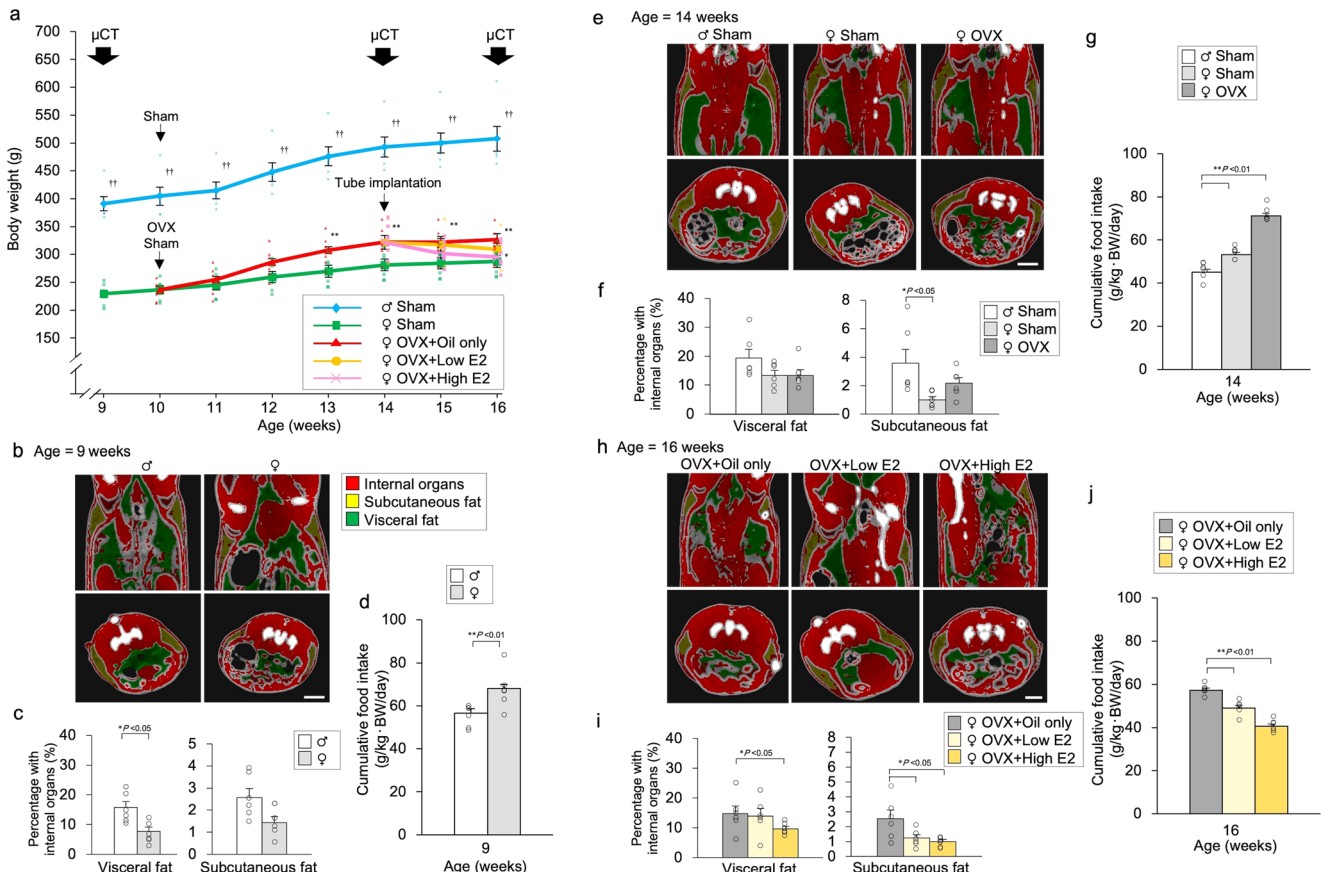

**Fig. 1 Relationship between body weight, fat mass, and feeding. a** Changes in body weight after treatment. Sham-operated male, sham-operated female, and ovariectomised (OVX) female rats at 10 weeks of age. Tube implantation [oil-only, low E2 (20 µg β-oestradiol/ml sesame oil), and high E2 (400 µg β-oestradiol/ml sesame oil)] was performed in OVX female rats at 14 weeks of age. The data are presented as the mean ± SEM (repeated-measures ANOVA) (**$^{**}P < 0.01$, compared with the treated rats; $^{††}P < 0.01$, compared with all female rats). **b** Micro-CT imaging (9 weeks of age) of visceral fat (green), subcutaneous fat (yellow), and internal organs (red) in male and female rats are shown. Scale bar indicates 1 cm. **c** Ratio of visceral and subcutaneous fat to that within internal organs in male and female rats are shown. The data are presented as the mean ± SEM (one-way ANOVA) ($^*P < 0.05$, compared with visceral fat of male rats). **d** Cumulative food intake for one day at 9 weeks of age. The data are presented as the mean ± SEM (t-test) ($^{**}P < 0.01$, compared with male rats). **e** Micro-CT imaging (14 weeks of age) of visceral fat (green), subcutaneous fat (yellow), and internal organs (red) in sham-operated male, sham-operated female, and OVX female rats are shown. Scale bar indicates 1 cm. **f** Ratio of visceral and subcutaneous fat to that within internal organs in sham-operated male, sham-operated female, and OVX female rats are shown. The data are presented as the mean ± SEM (one-way ANOVA) ($^*P < 0.05$, compared with sham-operated female rats). **g** Cumulative food intake for one day at 14 weeks of age. The data are presented as the mean ± SEM (one-way ANOVA) ($^{**}P < 0.01$, compared with sham-operated male rats). **h** Micro-CT imaging (16 weeks of age) of visceral fat (green), subcutaneous fat (yellow), and internal organs (red) in OVX + oil-only, OVX + low E2, and OVX + high E2 female rats are shown. Scale bar indicates 1 cm. **i** Ratio of visceral and subcutaneous fat to that within internal organs in OVX + oil-only, OVX + low E2, and OVX + high E2 female rats are shown. The data are presented as the mean ± SEM (one-way ANOVA) ($^*P < 0.05$, compared with OVX + oil-only female rats). **j** Cumulative food intake for one day at 16 weeks of age. The data are presented as the mean ± SEM (one-way ANOVA) ($^{**}P < 0.01$, compared with OVX + oil-only female rats).

PP (Fig. 4a). Interestingly, the high-dose E2 group demonstrated higher levels of OXT-mRFP1 fluorescence in the hypothalamus (SON, apPVN, dpPVN, and mPVN) compared to the low-dose group (Fig. 4b). This represents the first evidence that oestrogen regulates OXT expression in hypothalamic OXT neurons in a dose-related manner.

**Correlation between plasma oestrogen and OXT-mRFP1 fluorescence**. Among the OVX groups, the rats who received high and low E2 supplementation demonstrated significant changes in plasma E2 concentration (Fig. 4c). We investigated the correlation between plasma E2 and OXT-mRFP1 fluorescence in the hypothalamus (SON, apPVN, dpPVN, and mPVN) and PP. The results demonstrated that plasma E2 concentration significantly correlated with OXT-mRFP1 fluorescence in the SON, mPVN, and PP. Plasma E2 concentration was positively associated with OXT-mRFP1 fluorescence in the hypothalamus

(SON, apPVN, dpPVN, and mPVN) and PP, and it had no significant association with apPVN and dpPVN (Fig. 4c). After fixation with 4% paraformaldehyde, rats were used to investigate OXT-mRFP1 fluorescence. Plasma OXT was not measured because it fluctuated with anaesthesia.

**Effects of oestrogen replacement on *OXT* gene expression**. We performed OVX in 10-week-old female OXT-mRFP1 transgenic rats, conducted hormone replacement (oil only, low E2, and high E2) on week 14, and performed experiments on week 15. We investigated whether oestrogen affects *OXT* and *mRFP1* gene expression in decapitated rats (Exp. E). Among the OVX groups, the rats in the groups with low and high E2 supplementation demonstrated significant changes in plasma E2 and OXT concentration (Fig. 5a). In addition, plasma E2 concentration was significantly and positively correlated with plasma OXT concentration (Fig. 5b). The expression of *OXT* mRNA in the SON,

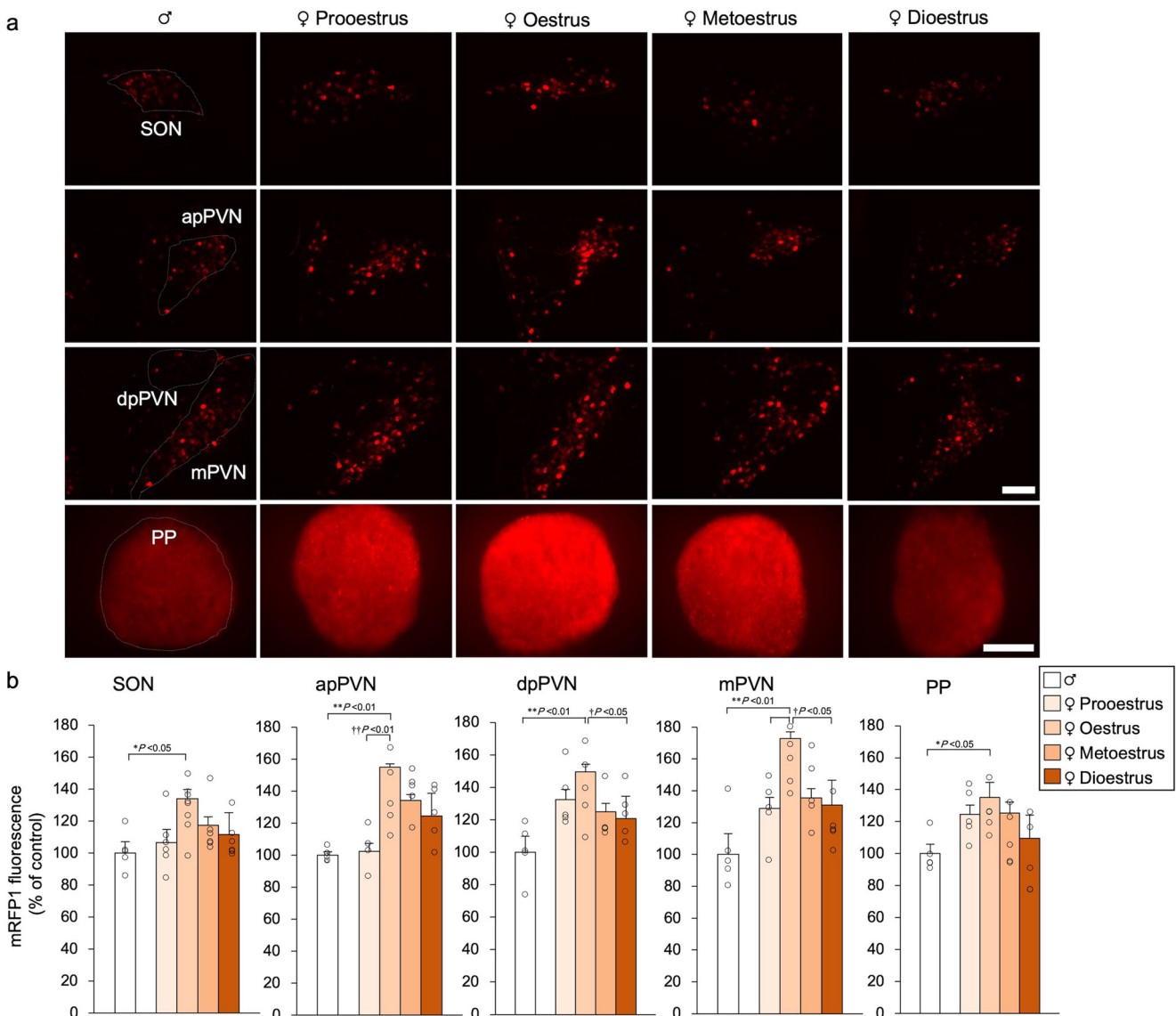

**Fig. 2 OXT-mRFP1 fluorescence in female and male rats. a** OXT-mRFP1-positive neurons fluoresce in the SON, apPVN, dpPVN, and mPVN of male, pro-oestrus female, oestrus female, metoestrus female, and dioestrus female OXT-mRFP1 transgenic rats (first three rows). Scale bar indicates 100 μm. OXT-mRFP1 fluorescence in the PP of male, pro-oestrus, oestrus, metoestrus, and dioestrus female OXT-mRFP1 transgenic rats (last row). Scale bar indicates 500 μm. **b** OXT-mRFP1 expression in the SON, apPVN, dpPVN, mPVN, and PP of the rats (male [$n = 5$], pro-oestrus female [$n = 5$], oestrus female [$n = 9$], metoestrus female [$n = 6$], dioestrus female [$n = 6$]). The data are presented as the mean ± SEM (one-way ANOVA) ($^{**}P < 0.01$, $^{*}P < 0.05$ when comparing females with male rats; $^{†}P < 0.05$ when comparing females in the pro-oestrous, metoestrous, and dioestrous stages with oestrus female rats).

dpPVN and mPVN was significantly increased in the rats with supplementation of high E2 compared with rats in the oil only group (Fig. 5c). Regarding *OXT* mRNA probe binding affinity in the SON, dpPVN, and mPVN, we found a statistical difference in the OVX + oil only and OVX + high E2 groups (Fig. 5d). The high-dose E2 group demonstrated higher levels of *OXT* mRNA in the dpPVN than the low-dose group (Fig. 5d). We investigated the correlation between plasma E2, plasma OXT, and changed in *OXT* mRNA levels in the hypothalamus (SON, dpPVN, and mPVN). The results demonstrated that plasma E2 concentration significantly and positively correlated with *OXT* mRNA levels in the hypothalamus (SON, dpPVN, and mPVN) (Fig. 5e). Plasma OXT concentration was significantly and positively related to *OXT* mRNA in the dpPVN and mPVN (Fig. 5f). This indicates that oestrogen regulates the hypothalamic *OXT* gene in a dose-related manner.

**Effects of oestrogen replacement on *mRFP1* gene expression.** The expression of *mRFP1* mRNA in the SON, dpPVN and mPVN was significantly increased in the rats with high E2 supplementation than in the rats in the oil-only group (Fig. 6a). There was also a statistically significant difference in *mRFP1* mRNA probe binding affinity in the SON, dpPVN, and mPVN between the OVX + oil-only and OVX + high E2 groups (Fig. 6b). The high-dose E2 group demonstrated higher levels of *OXT* mRNA in the dpPVN than the low-dose group (Fig. 6b). We investigated the correlation between plasma E2, plasma OXT, and changes in *mRFP1* mRNA levels in the hypothalamus (SON, dpPVN, and mPVN). The results demonstrated that plasma E2 concentrations were significantly and positively correlated with *mRFP1* mRNA in the dpPVN and mPVN (Fig. 6c). Plasma OXT concentration was also significantly positively related to *mRFP1* mRNA in the dpPVN and mPVN (Fig. 6d). These results

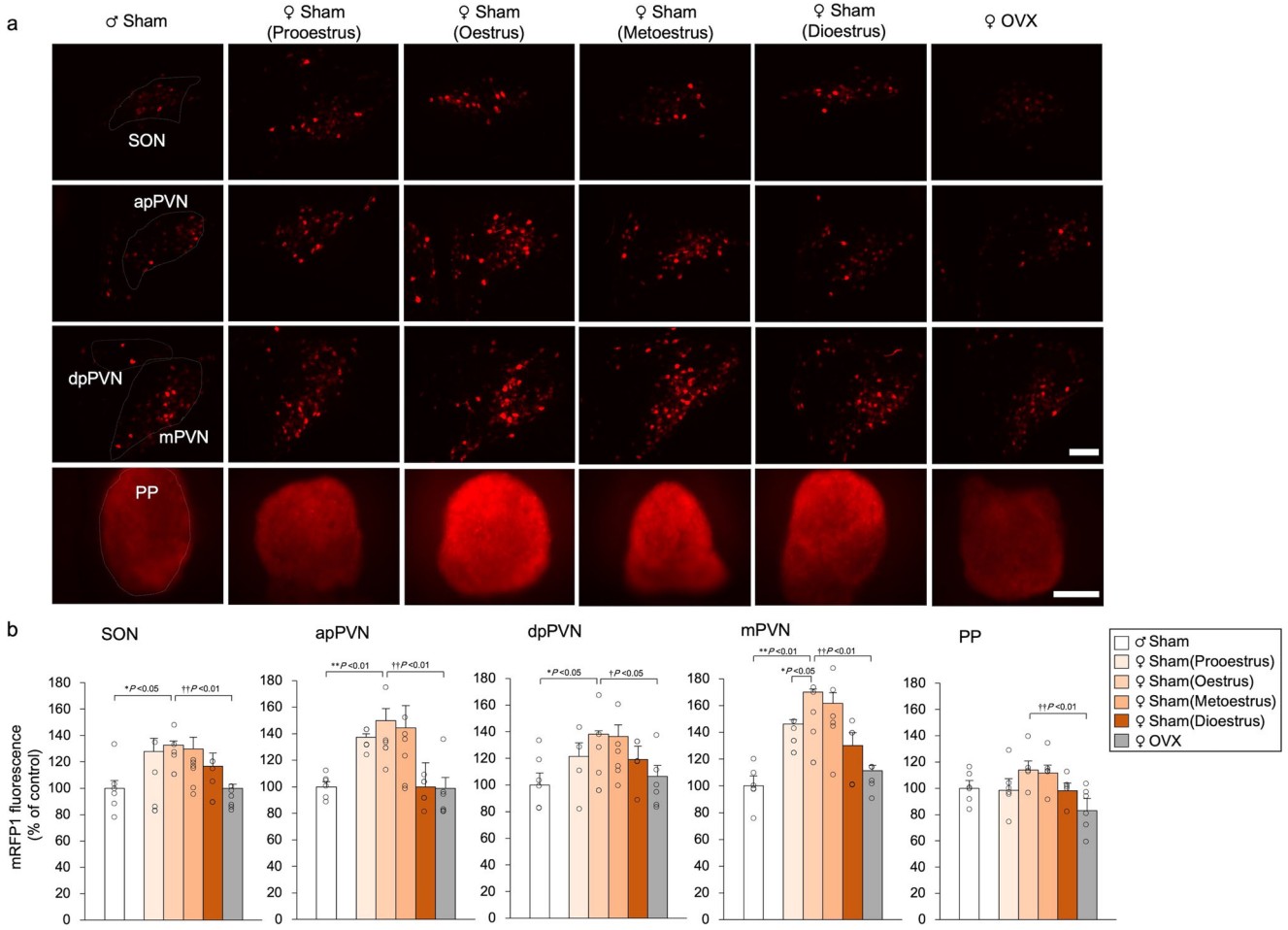

**Fig. 3 Effects of OVX on OXT-mRFP1 fluorescence. a** OXT-mRFP1-positive neurons fluoresce in the SON, apPVN, dpPVN, and mPVN of sham-operated male, sham-operated female in the pro-oestrus, oestrus, metoestrus, and dioestrus stages, and OVX female OXT-mRFP1 transgenic rats (first three rows). Scale bar indicates 100 µm. OXT-mRFP1 fluorescence in the PP of sham-operated male, sham-operated female in the pro-oestrus, oestrus, metoestrus, and dioestrus stages, and OVX female OXT-mRFP1 transgenic rats (last row). Scale bar indicates 500 µm. **b** OXT-mRFP1 expression in the SON, apPVN, dpPVN, mPVN, and PP of the rats (male [$n = 5$], pro-oestrus female [$n = 5$], oestrus female [$n = 5$], metoestrus female [$n = 5$], dioestrus female [$n = 5$], and OVX female [$n = 6$]). The data are presented as the mean ± SEM (one-way ANOVA) ($**P < 0.01$, $*P < 0.05$, compared with male rats; $††P < 0.01$, $†P < 0.05$, compared with oestrus and OVX female rats).

indicates that oestrogen regulates the hypothalamic OXT-*mRFP1* gene in a dose-dependent manner.

**Effect of peripheral administration of CCK-8 on food intake with oestrogen replacement**. We performed CCK-8 administration experiments to investigate the relationship between oestrogen and food intake. All Wistar female rats with OVX were divided into four groups: oil only and high-dose oestrogen in the subcutaneous tube with i.p. administration of saline or CCK-8 (Exp. F). Rats receiving high doses of oestrogen experienced significant weight loss (Fig. 7a). Rats supplemented with high doses of oestrogen consumed significantly less food throughout the day. Cumulative food intake was significantly decreased in the OVX + high E2 group compared to the OVX + oil only group (Fig. 7b). CCK-8 was administered to examine the amount of food consumed. Cumulative food intake was significantly decreased at 0.5 h, 1 h, and 1.5 h after i.p. administration of CCK-8. There was a significant difference between the OVX + oil only and OVX + high E2 groups at 1.5 h after i.p. administration of saline and at 3 h after i.p. administration of CCK-8. After 6 h, there was no significant difference in cumulative food intake among all groups (Fig. 7c).

**Effect of oestrogen on Fos expression in OXT-ir neurons**. We conducted immunohistochemistry to assess the levels of Fos and OXT in the hypothalamus (Exp. G). All Wistar female rats with OVX were divided into two groups: oil only and high-dose oestrogen tubing. Tissues were harvested and evaluated for the expression of Fos and OXT via double-fluorescence immunohistochemistry (FIHC). We quantified immunofluorescently-labelled OXT+, Fos+, and OXT + /Fos+ double-labelled cells in the SON and PVN at 1.5 h after i.p. administration of CCK-8. (Fig. 7d). The number and percentage of OXT+/Fos+ cells were significantly higher when CCK-8 was administered than when saline was administered. Among these rats, the number and percentage of OXT+/Fos+ cells were significantly higher in the OVX + high E2 group than in the OVX + oil only group (Fig. 7e).

**Effect of pre-treatment with OXTR-A on food intake**. In the previous experiment (Exp. F), there was a significant difference in food intake between the OVX + oil-only and OVX + high E2 groups at 3 h after i.p. administration of CCK-8. Therefore, we assessed food intake for 3 h after i.p. administration of CCK-8 and i.c.v. administration of OXTR-A. All Wistar female rats with OVX were divided into four groups: oil only and high-dose

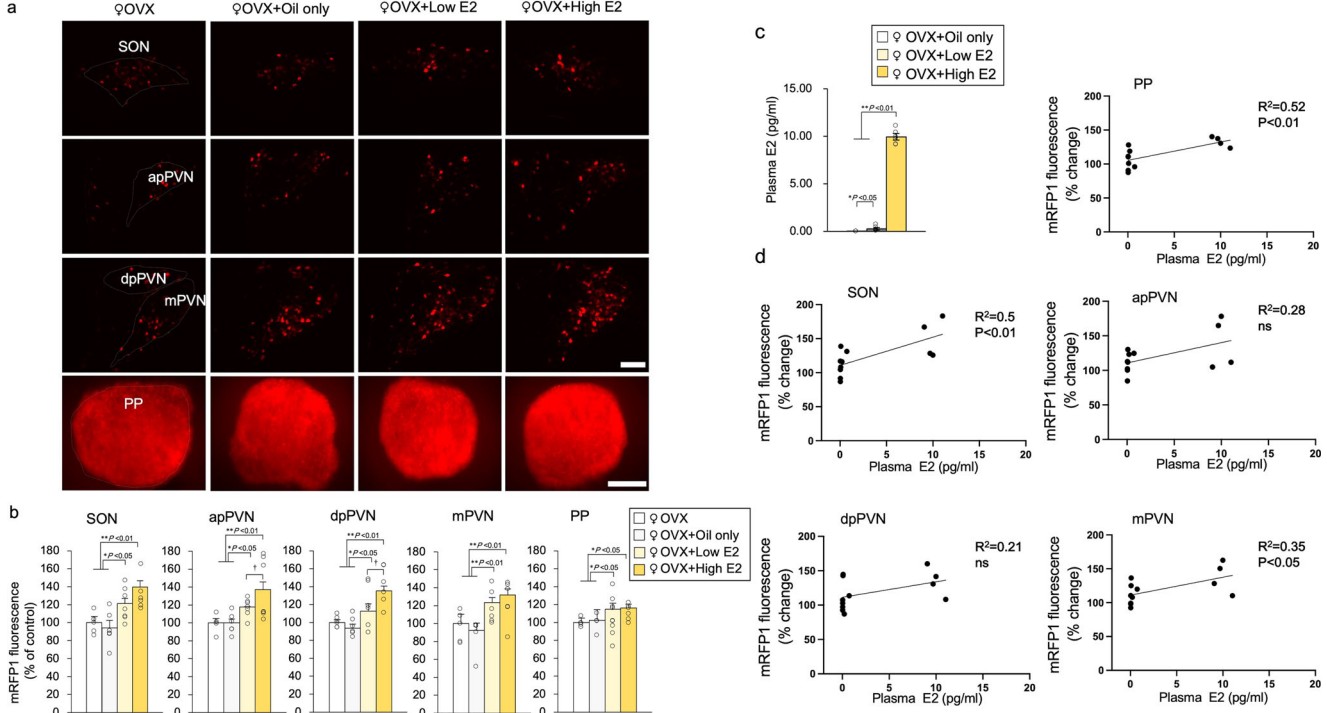

**Fig. 4 Effects of oestrogen replacement on OXT-mRFP1 fluorescence. a** OXT-mRFP1 positive neurons fluoresce in the SON, apPVM, mPVN, and dpPVN of female OXT-mRFP1 transgenic rats in the OVX group, OVX + oil-only group, OVX + low E2 (20 μg β-oestradiol/ml sesame oil) group, and OVX + high E2 (400 μg β-oestradiol/ml sesame oil) groups, respectively (first three rows). Scale bar indicates 100 μm. OXT-mRFP1 fluorescence in the PP of female OXT-mRFP1 transgenic rats in the OVX group, OVX + oil-only group, OVX + low E2 group, and OVX + high E2 groups (last row). Scale bar indicates 500 μm. **b** OXT-mRFP1 expression of the SON, apPVN, dpPVN, mPVN, and PP of the rats (OVX [$n = 6$], OVX + oil-only [$n = 5$], OVX + low E2 [$n = 6$] and OVX + high E2 [$n = 6$]). The data are presented as the mean ± SEM (one-way ANOVA) ($^{**}P < 0.01$, $^{*}P < 0.05$, compared with OVX group; $^{†}P < 0.05$, compared with OVX + low E2 and OVX + high E2 groups). **c** Plasma E2 concentrations (OVX + oil-only [$n = 4$], OVX + low E2 [$n = 4$] and OVX + high E2 [$n = 4$]). The data are presented as the mean ± SEM (one-way ANOVA) ($^{**}P < 0.01$, compared with OVX + high E2 groups; $^{*}P < 0.05$, compared with OVX + low E2 groups). **d** Regression analysis for plasma E2 and mRFP1 fluorescence in the PP, SON, apPVN, dpPVN and mPVN of the rats. The regression line and probability value for the slope are shown. The statistical significance of the slope was set at $P < 0.05$.

oestrogen in the subcutaneous tube with i.p. administration of CCK-8 and i.c.v. administration of saline or OXTR-A (Exp. H). Rats receiving high doses of oestrogen experienced significant weight loss (Fig. 8a). Cumulative food intake for 3 h was significantly increased in the i.c.v. administration of OXTR-A group compared to the i.c.v. administration of saline group (Fig. 8b).

## Discussion

The present study provides the first evidence that hypothalamo-neurohypophysial OXT is oestrogen-dependent and shows dynamic changes during the oestrus cycle. The novelty of this study was the success in identifying oestrogen-dependent hypothalamic-pituitary OXT changes by observing the fluorescent intensities of mRFP1 in OXT neurons and their axon terminals in the PP. OXT-mRFP1 fluorescence intensity in the SON and PVN was expressed most strongly among adult oestrous female rats and was significantly reduced in OVX rats. Oestrogen supplementation restored fluorescence intensity and *OXT-mRFP1* mRNA expression levels in the SON and PVN in OVX rats in a dose-related manner. Thus, the dynamics of hypothalamic OXT expression is regulated by oestrogen. The advantage of this study was the quantitative demonstration of the hypothalamic-pituitary OXT system by examining OXT blood concentrations, correlation coefficient and *OXT* mRNA expression levels in the SON and PVN. We confirmed that feeding suppression induced by the peripheral administration of CCK-8 resulted in a difference in the activation of OXT neurons, and this was enhanced among

oestrogen-replaced OVX female rats. Therefore, we investigated changes in food intake when CCK-8, oestrogen, and OXTR-A administration are combined and proved the relationship between oestrogen and OXT.

OXT is mainly produced in neurosecretory neurons located in the SON and PVN in the hypothalamus. We successfully generated transgenic rats bearing an OXT-mRFP1 fusion gene, which enabled the visualisation of OXT expression[21,22,27–30]. The PVN is divided into regions such as the apPVN, dpPVN, and mPVN, and previous studies on OXT-mRFP1 transgenic rats reported different findings[31]. OXT+ neurons in the SON and mPVN project their axons to the PP, where OXT is thereby secreted into the systemic circulation and elicits activity peripherally[32]. There are reports that in males, OXT is involved in sexual behaviour, ejaculation, and transport of spermatozoa. In females, the peripheral effects of OXT are related to labour and lactation[33]. With respect to the central functions, OXT is also somatodendrically released from neurons in the SON and mPVN and acts directly on the brain[34]. Neurons expressing OXT receptors are ubiquitous in the brain and have a wide range of functions[35]. It has been reported that OXT is not only associated with confidence and bond formation but is also strongly associated with autism[36,37]. In the present study, the observed differences related to the sex of OXT-mRFP1 rats suggest that the production of OXT in the hypothalamus differs according to sex. This may further indicate that the central actions of OXT can also vary with sex.

A pathway that projects OXT from the apPVN and dpPVN to the medulla and spinal cord has been identified[31]. This pathway

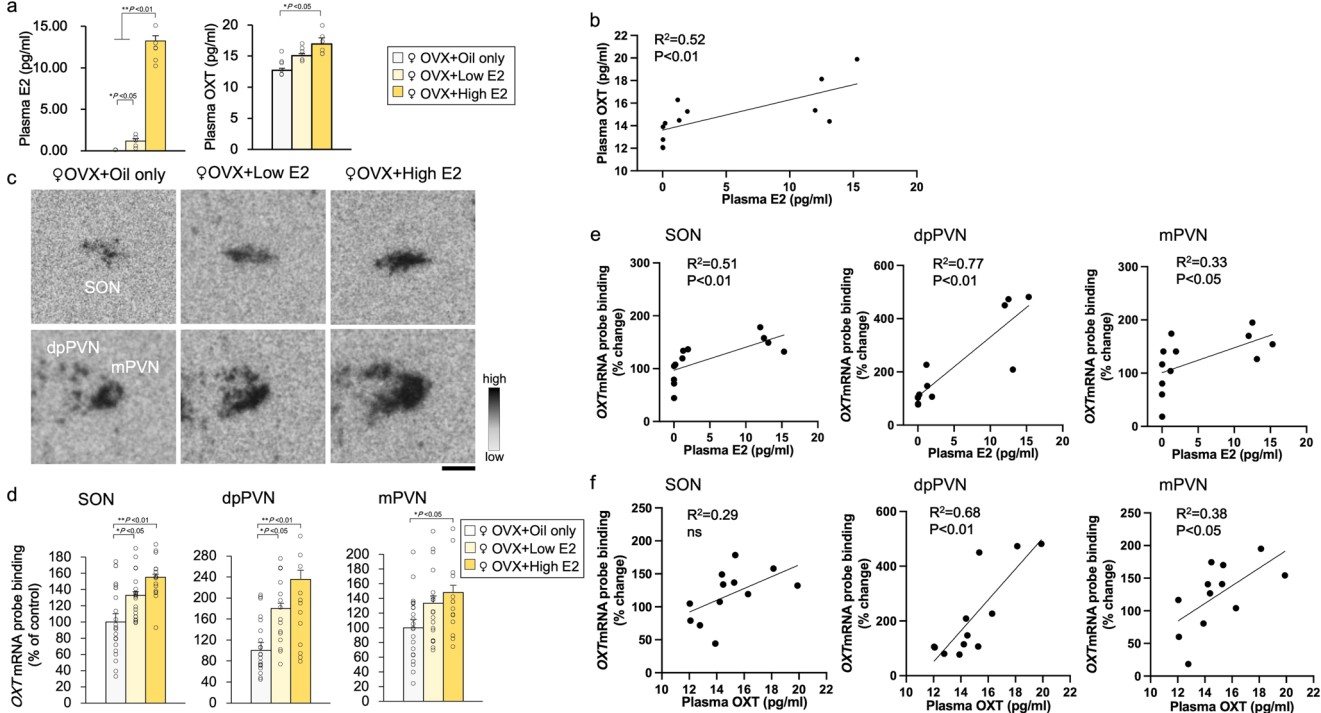

**Fig. 5 Effects of oestrogen replacement on OXT gene expression. a** Plasma E2 and OXT concentrations (OVX + oil-only [$n = 4$], OVX + low E2 [$n = 4$] and OVX + high E2 [$n = 4$]). The data are presented as the mean ± SEM (one-way ANOVA) (**$P < 0.01$, *$P < 0.05$, compared with OVX + oil-only group). **b** Regression analysis for plasma E2 and OXT. The regression line and probability value for the slope are shown. The statistical significance of the slope was set at $P < 0.05$. **c** Representative images hybridised with a 35S-labelled oligodeoxynucleotide probe complementary to *OXT* in the SON, dp PVN, and mPVN of female OXT-mRFP1 transgenic rats in the OVX + oil-only ($n = 6$), OVX + low E2 ($n = 6$), and OVX + high E2 ($n = 6$) groups, respectively. Scale bar indicates 100 µm. **d** *OXT* mRNA of the SON, dpPVN and mPVN of the rats (OVX + oil-only, OVX + low E2, and OVX + high E2). The data are presented as the mean ± SEM (one-way ANOVA) (**$P < 0.01$, *$P < 0.05$, compared with OVX + oil-only group). **e** Regression analysis for plasma E2 and changes in *OXT* mRNA levels in the SON, dpPVN, and mPVN. The regression line and probability value for the slope are shown. The statistical significance of the slope was set at $P < 0.05$. **f** Regression analysis for plasma OXT and changes in *OXT* mRNA levels in the SON, dpPVN, and mPVN. The regression line and probability value for the slope are shown. The statistical significance of the slope was set at $P < 0.05$.

has been reported to induce analgesic effects and regulate pain via the autonomic nervous system, and to modulate gastrointestinal motility[38] and cardiovascular responses[39]. There are clinical reports that the pain threshold is high during pregnancy and postpartum[40]. Chronic pain after caesarean section is reported to be a tenth as severe as that after other open surgeries[41]. We speculate that these observations could be explained by hypothalamic OXT dynamics. In our study, OXT-mRFP1 expression was increased in the apPVN and dpPVN throughout the oestrus cycle, and this may, in turn, regulate pain thresholds.

Spontaneous ovulator animals have an ovulation cycle. Humans have an oestrus cycle of approximately 28 days while rats have a shorter cycle of 4 5 days. The blood levels of hormones secreted from the ovaries also fluctuate periodically, and these dynamics are common between humans and rats. Oestrogen acts on the mucous membrane epithelium and changes its histology. Therefore the oestrus cycle can be monitored through a vaginal liquid smear examination[42]. Based on this examination, the oestrus cycle in rats can be classified into the pro-oestrus, oestrus, metoestrus, and dioestrus cycles. The highest increase in blood oestrogen levels occurs when the luteinizing hormone (LH) surge coincides with the pro-oestrus stage. Ovulation subsequently occurs several hours later[43]. In OXT-mRFP1 transgenic rats, the fluorescence intensity of mRFP1 has been shown to be delayed by several hours after stimulation[21,31,44]. It is thought that the fluorescence intensity of mRFP1 increases several hours after the point of the highest pro-oestrus production of oestrogen by the

ovary. This, therefore, indicates that fluorescence intensity peaks during the oestrus stage.

In females, oestrogen levels are reduced as a result of age-related reductions in ovarian function and when the ovaries are removed due to gynaecological surgery or treatment. In addition to mammary glands and genital organs, oestrogen acts on the liver, cardiovascular system, bones, and the brain[13,45,46]. Therefore, ovarian dysfunction can elicit various symptoms. In this study, we mimicked ovarian dysfunction/menopause through ovariectomies. We observed that OVX reduced mRFP1 fluorescence in the SON and PVN (ap, dp, m). Thus, OXT production may have been decreased in all hypothalamic areas. Decreased OXT production suggests that broad-ranging central OXT effects may be attenuated by OVX.

There are three known variants of oestrogen—oestrone (E1), oestradiol (E2), and oestriol (E3), and three known subtypes of ERs—ERα, ERβ, and G protein-coupled receptor 30[47]. ERs are expressed systemically, and ERα and ERβ are localised in the brain. However, only ERβ has been reported to be expressed in the PVN and SON in the hypothalamus[12]. E2, which elicits the strongest effects, is often used as an experimental or therapeutic drug. In this experiment, the type of oestrogen used was E2. Oestrogen replacement has various effects on the liver, cardiovascular system, bones, and the brain. Notably, it is used to treat menopausal symptoms[48–50]. The subcutaneous administration of oestrogen has fewer side effects compared to intravenous and oral administration and can reproduce the systemic effects of oestrogen[51]. In our study, the subcutaneous administration of E2

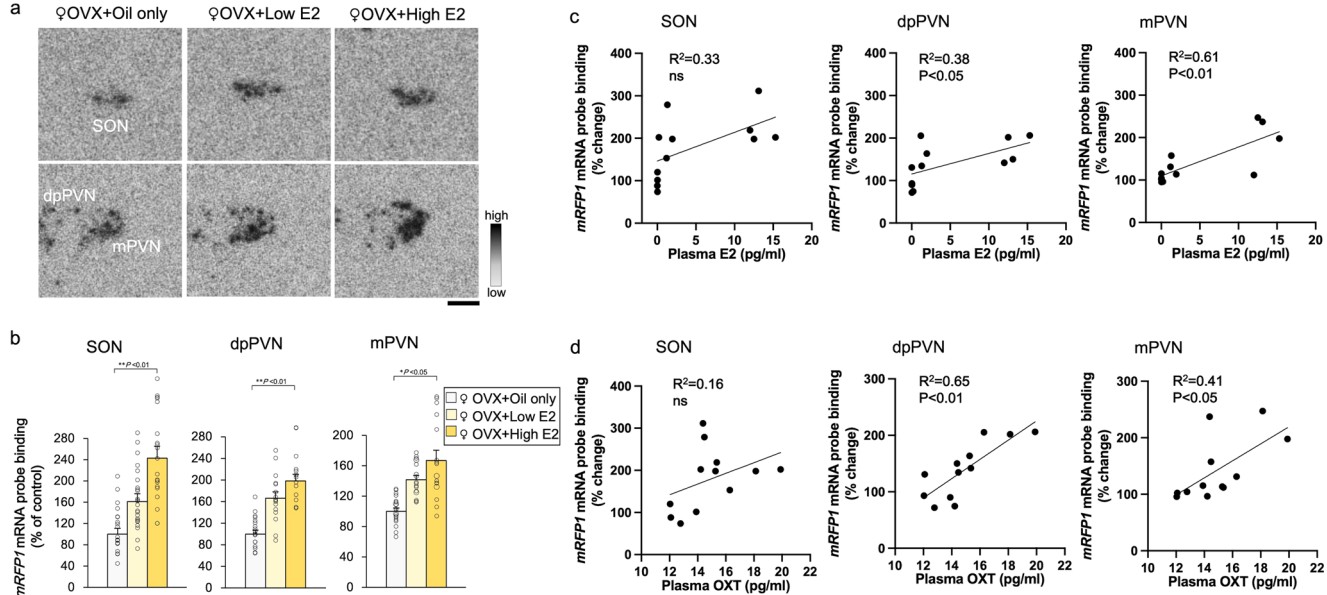

**Fig. 6 Effects of oestrogen replacement on mRFP1gene expression. a** Representative images hybridised with a 35S-labelled oligodeoxynucleotide probe complementary to *mRFP1* in the SON, dpPVN, and mPVN of female OXT-mRFP1 transgenic rats in the OVX + oil-only ($n = 6$), OVX + low E2 ($n = 6$), and OVX + high E2 ($n = 6$) groups, respectively. The scale bar indicates 100 μm. **b** *mRFP1* mRNA in the SON, dpPVN, and mPVN of the rats (OVX + oil-only, OVX + low E2, and OVX + high E2 groups). The data are presented as the mean ± SEM (one-way ANOVA) (**$P < 0.01$, *$P < 0.05$, compared with OVX + oil only group). **c** Regression analysis for plasma E2 and changes in *mRFP1* mRNA levels in the SON, dpPVN, and mPVN. The regression line and probability value for the slope are shown. The statistical significance of the slope was set at $P < 0.05$. **d** Regression analysis for plasma OXT and changes in *mRFP1* mRNA levels in the SON, dpPVN, and mPVN. The regression line and probability value for the slope are shown. The statistical significance of the slope was set at $P < 0.05$.

in OVX female rats restored mRFP1 fluorescence intensity in the SON and PVN (ap, dp, m) in a dose-related manner. Thus, the subcutaneous administration of E2 may increase OXT production in all hypothalamic regions.

Oestrogen has further been reported to suppress feeding. Mechanistically, oestrogen has been shown to increase pro-opiomelanocortin gene expression, which induces appetite suppression via the signal transducer and activator of transcription 3 in the hypothalamus[52]. Oestrogen also regulates hypothalamic OXT activity in the solitary tract nucleus and thereby suppresses food intake[19]. Furthermore, i.c.v. OXT administration reportedly reduces food intake significantly at all stages of the oestrous cycle except the oestrous cycle, suggesting that female hormones may regulate the feeding effects of OXT[53]. Similar to the previous reports, we demonstrated that oestrogen administration suppressed feeding and promoted weight loss. Furthermore, we confirmed that hypothalamic OXT production increased depending on oestrogen levels. This suggests that food consumption may differ based on oestrogen levels.

Oestrogen and OXT share a common antifeedant activity[22]. Food consumption was measured following the administration of CCK-8, which is known to selectively activate OXT neurons[23]. We demonstrated the link between OXT and oestrogen by counting Fos+/OXT+ neurons. Fos is an early expression pro-oncogene expressed at low levels in most cell types[54] and can be activated by various second messenger signals. However, Fos is an indicator of cell activation. Accordingly, the evaluation of Fos expression cannot be used to determine any direct effects related to feeding versus other secondary effects, which may likewise induce cellular activation. The gastrointestinal hormone CCK-8 acts on OXT neurons in the hypothalamus through the solitary tract nucleus to increase blood OXT concentrations and suppress appetite[23]. We demonstrated that food consumption was lowest when CCK-8 was administered alongside oestrogen

supplementation. Thus, we demonstrated that oestrogen could further induce OXT production when CCK-8 was administered. Likewise, the strongest levels of appetite suppression corresponded with the highest proportions of OXT+/Fos+ neurons. These results suggest that oestrogen administration may enhance hypothalamic OXT production clinically.

Various studies have demonstrated the efficiency of oestrogen and OXT as anti-obesity peptides[55]. Intracerebroventricular or peripheral (i.p. and subcutaneous) injection of OXT decreases food intake, body weight, and fat mass in rats and mice[39,56]. In the present study, fat was divided into visceral fat and subcutaneous fat, with differences between sexes. The percentage of visceral fat was higher in males, and subcutaneous fat was altered in females. In this experiment, oestrogen was associated with fat composition. However, it is unclear whether OXT was directly involved.

Oestrogen, OXT and CCK-8 decreases food intake. CCK-8 acts on OXT neurons in the hypothalamus to decreases appetite. Oestrogen, OXT, and CCK-8 have been known to suppress feeding, but no studies combining them were conducted. In this study, the addition of oestrogen to CCK-8 resulted in a more marked decrease in food intake and activation of OXT neurons. Oestrogen was also found to regulate OXT neurons in transgenic rats via in situ experiments. The oestrogen receptor ERβ is localised in the PVN and SON of the hypothalamus[12]. Therefore, oestrogen may act on these hypothalamic areas via ERβ to activate OXT neurons.

To the best of our knowledge, this is the first report demonstrating a difference in the dynamics of hypothalamic OXT in rats based on the sex. Additionally, we demonstrated that hypothalamic OXT expression is specifically dependent upon oestrogen, as E2 administration increased central OXT production in OVX rats. However, our study has several limitations. First, we did not assess any associations among hypothalamic OXT, the OXT

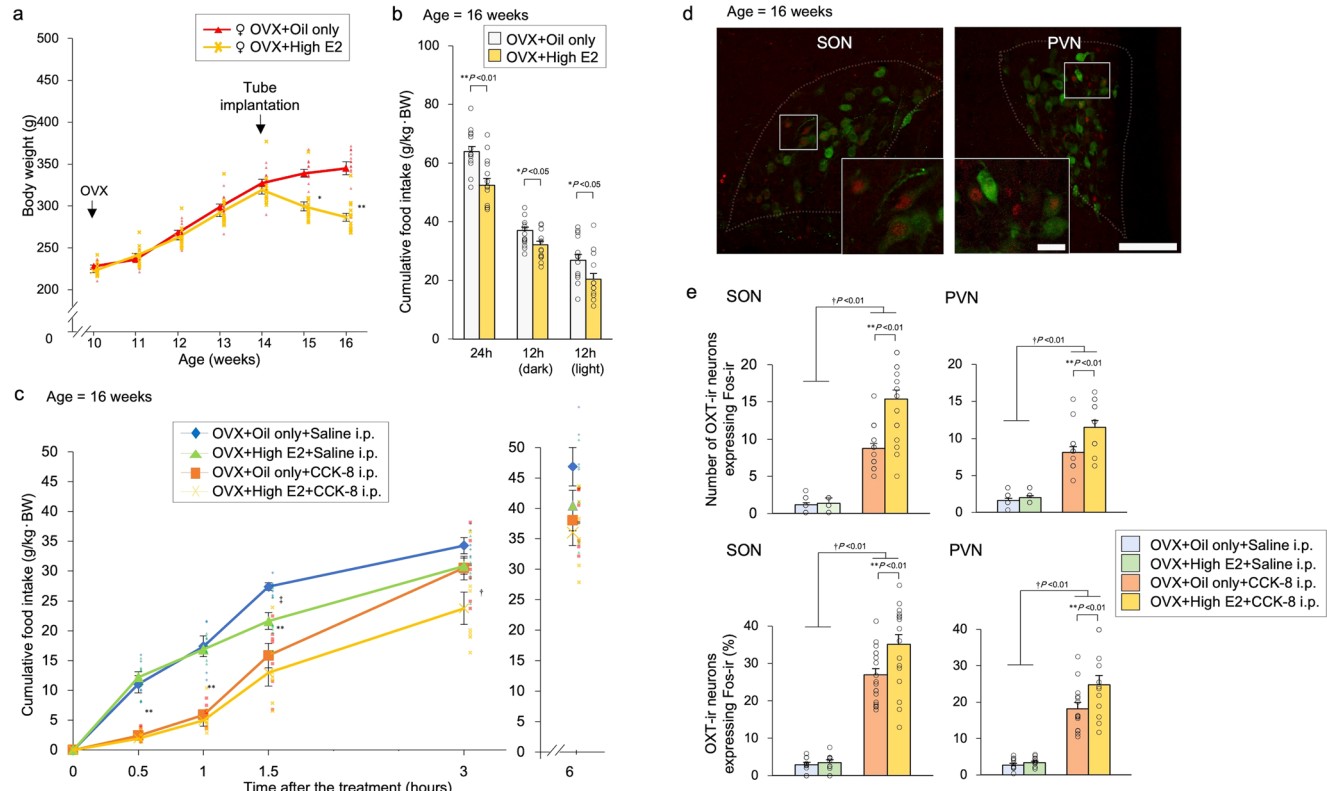

**Fig. 7 Effect of peripheral administration of CCK-8 on food intake with oestrogen replacement. a** Changes in body weight after treatment. Tube implantation (oil only and high E2) was performed in OVX female rats at 14 weeks of age. The data are presented as the mean ± SEM (repeated-measures ANOVA) (**$P < 0.01$, *$P < 0.05$, compared with OVX + oil only and OVX + high E2 groups). **b** Cumulative food intake for one day, dark-time, and light-time. The data are presented as the mean ± SEM (one-way ANOVA) (**$P < 0.01$, *$P < 0.05$, compared with OVX + oil only and OVX + high E2 groups). **c** Cumulative food intake at 0.5 h, 1 h, 1.5 h, 3 h, and 6 h after i.p. administration of saline or CCK-8 (50 µg/kg body weight). The data are presented as the mean ± SEM (repeated-measures ANOVA) (**$P < 0.01$ compared with saline and CCK-8 i.p. groups; †$P < 0.05$, compared with OVX + oil only and OVX + high E2 groups; ‡$P < 0.05$, compared with OVX + oil only and OVX + high E2 groups). **d** Photomicrographs of the SON and PVN obtained during double-FIHC of Fos and OXT 1.5 h after i.p. administration of CCK-8 to the rats in the OVX + high E2 group. Representative images of Fos-immunoreactive (ir) expression are shown as red-coloured cells, and OXT-ir expression is shown as green cytoplasmic cells in the panel. Merged images of Fos-ir and OXT-ir neurons in the SON and PVN are shown. Scale bars in the low-power and high-power photomicrographs indicate 100 µm and 20 µm, respectively. **e** The number of OXT-ir neurons expressing Fos-ir in the SON and PVN 1.5 h after i.p. administration of saline or CCK-8 to the rats in the OVX + oil only group or OVX + high E2 group. The average percentage of OXT-ir neurons expressing Fos-ir in the SON and PVN 1.5 h after i.p. administration of saline or CCK-8 to the rats in the OVX + oil only group or OVX + high E2 group. The data are presented as the mean ± SEM (two-way ANOVA) (**$P < 0.01$, compared with saline and CCK-8 i.p. groups; †$P < 0.05$, compared with OVX + oil only and OVX + high E2 groups).

receptor, and sex hormones other than oestrogen (*e.g.*, progesterone). Second, all hypothalamic examinations were conducted in rats and not clinically in humans. Finally, no adverse effects of oestrogen replacement were considered.

## Methods

**Animals**. All rats were treated after 10 weeks of age once the oestrus cycle was established. Adult male and female OXT-mRFP1 Wistar transgenic rats (aged 10 16 weeks and weighing 223–474 g) were bred and maintained under normal laboratory conditions (12-h light, 12-h dark cycle) with free access to food and drinking water[21,22]. The OXT-mRFP1 transgenic rats were created by inserting the *mRFP1* gene into the *OXT* gene. This reporter strain facilitates the visualisation of OXT dynamics in the hypothalamus through the quantification of OXT fluorescence intensity changes under various stimulation loads[30,31]. All rats were genotypically screened through polymerase chain reaction analysis of their genomic DNA extracted via ear biopsies[21]. Adult female Wistar rats (aged 15 16 weeks and weighing 271–337 g) were used in experiment 4 (below). All rats were housed as one or three per plastic cage (transparent polymethylpentene; TR-TPX-200A, Tokiwa Kagaku Kikai, Tokyo, Japan) in an air-conditioned room (22–25 °C) with a 12-h light cycle (7:00 A.M. to 7:00 P.M.) and ad libitum access to food (CLEA Rodent Diet CE-2; CLEA Japan, Tokyo, Japan) and water. All transgenic rats and Wistar rats delivered at the same time were used for each experiment. All experiments were performed in strict accordance with guidelines on the use and care of laboratory animals as set forth by the Physiological Society of Japan and

approved (No.AE10-012) by the Ethics Committee of Animal Care and Experimentation of the University of Occupational and Environmental Health, Japan.

**Surgical procedures**. Bilateral ovariectomies were conducted on rats to induce an oestrogen-deficient state. Ovariectomies were conducted by opening the flanks of the rat by ~1 cm and removing the ovaries attached to the end of the double-horned uterus. In sham operations, the abdominal cavity was closed without treatment. Rats undergoing OVX and sham operations were anesthetised with an i.p. injection of a cocktail of three different anaesthetic agents (0.3 mg/kg of medetomidine, 4.0 mg/kg of midazolam, and 5.0 mg/kg of butorphanol).

For hormone replacement, hormone-containing tubes were subcutaneously implanted into in the mid-back region of the rats. Rats were anesthetised with isoflurane (3% isoflurane with a flow rate of 5.0 L/min). Silastic tubing (1.57 mm inner diameter; 3.18 mm outer diameter; 37.0 mm in length; Dow Corning, Midland, MI, USA) was filled with fat-soluble E2 (17β-oestradiol ≥ 98%; Sigma-Aldrich, Tokyo, Japan) dissolved in sesame oil (Sigma-Aldrich)[57,58].

For i.c.v. administration, animals were implanted with stainless steel canulae targeting the lateral ventricle. They were anaesthetised (i.p. injection of a cocktail of three different anaesthetic agents [0.3 mg/kg of medetomidine, 4.0 mg/kg of midazolam, and 5.0 mg/kg of butorphanol]) and placed in a stereotaxic frame. Stainless steel guide canulae (550 µm outer diameter and 10 mm length) were stereotaxically implanted at the following coordinates: 0.8 mm posterior to the bregma, 1.4 mm lateral to the midline, and 2.0 mm below the surface of the left cortex, such that canula tips were 1.0 mm above the left cerebral ventricle[59]. Two stainless steel anchoring screws and acrylic dental cement were used to secure the

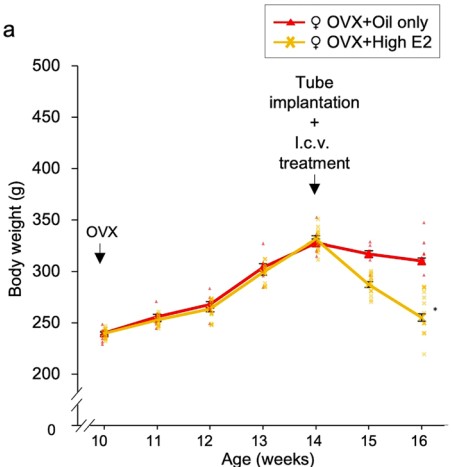
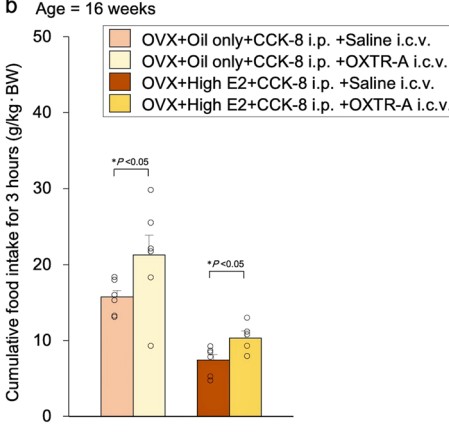

**Fig. 8 Effect of pre-treatment with OXT receptor antagonist (OXTR-A) on food intake. a** Changes in body weight after treatment. Tube implantation (OVX + oil only and OVX + high E2) and i.c.v. treatment was performed in OVX female rats at 14 weeks of age. The data are presented as the mean ± SEM (repeated-measures ANOVA) (*$P < 0.05$, compared with OVX + oil only and OVX + high E2 groups). **b** Cumulative food intake for 3 h after i.p. administration of CCK-8 (50 μg/kg body weight) and i.c.v. administration of saline or OXTR-A (150 ng/μl) to the rats in the OVX + oil only group or OVX + high E2 group. The data are presented as the mean ± SEM (two-way ANOVA) (*$P < 0.05$, compared with saline i.c.v. and OXTR-A i.c.v. groups).

cannulae in place. After the surgical procedure, animals were handled daily, individually housed in a plastic cage, and allowed to recover for at least 10 days.

**Test substances**. CCK-8 (Peptide Institute, Osaka, Japan) was dissolved in 0.9% sterile physiological saline (Otsuka Pharmaceutical Co., Ltd., Tokyo, Japan) to a concentration of 10 μg/ml. OXTR-A (L-368899, Tocris Bioscience, Bristol, UK) was dissolved in 0.9% sterile physiological saline (150 ng/μl).

**Micro-CT**. Rats were anesthetised with an i.p. injection of a cocktail of three different anaesthetic agents (0.3 mg/kg of medetomidine, 4.0 mg/kg of midazolam, and 5.0 mg/kg of butorphanol) before undergoing micro-CT scanning. Body fat images were acquired by a micro-CT system (CosmoScan GX; Rigaku, Tokyo, Japan) with a resolution of $148 \times 148 \times 148$ μm$^3$ (90 kVp, 88 μA, 555.33 ms integration time). The CT images of body fat were visualised using Analyze 12.0 software (AnalyzeDirect, Inc., KS, USA). The total fat volume in the body was measured from the base of the ensiform cartilage to the pelvic floor, and the fat volume was further distinguished into visceral and subcutaneous fat.

**Experimental procedure**
*Measurement of body weight, fat mass, and feeding.* We first aimed to ascertain the effects of sex and oestrogen on body weight, fat mass, and feeding. In the first experiment [(Exp. A), $n = 30$], we assessed body weight in male and female Wistar rats. Wistar rats were divided into five groups ($n = 6$ in each group): sham-operated male, sham-operated female, only OVX, OVX + low E2 (20 μg β-oestradiol/ml sesame oil) replacement, and OVX + high E2 (400 μg β-oestradiol/ml sesame oil) replacement groups. We performed sham operations and OVX in 10-week-old male and female rats, and conducted hormone replacement (tube implantation) with OVX rats at week 14. Micro-CT was performed at 9, 14, and 16 weeks of age to measure the ratio of visceral and subcutaneous fat to internal organs.

*Assessing differences in OXT-mRFP1 fluorescence in male and female transgenic rats.* We aimed to ascertain differences in OXT-mRFP1 fluorescence between 10-week-old female rats undergoing a normal oestrus cycle and 10-week-old male rats [second experiment (Exp. B), $n = 30$]. Female OXT-mRFP1 transgenic rats were further divided based on the four oestrus stages (pro-oestrus, oestrus, metoestrus, and dioestrus stages). In total, we assessed five groups (one male, four females; $n = 5$ 9 per group). The stage of the oestrus cycle was confirmed by examining vaginal smears collected every morning from the rats by two researchers. Briefly, the pro-oestrus stage is identified mainly by nucleated cells, oestrus stage by all keratinocyte cells, metoestrus stage by the presence of many white blood cells, and dioestrus stage by the presence of few white blood cells and other cells. Rats with irregular oestrus cycles were excluded from the experiment.

*Assessing the effects of OVX on OXT-mRFP1 fluorescence.* For the third experiment (Exp. C, $n = 33$), all OXT-mRFP1 transgenic rats were divided into six groups, including five sham-operated groups (comprising both males and females at all oestrus cycle stages) and one OVX group ($n = 5$ 6 per group). We confirmed the oestrogen-deficient state (e.g., dioestrus stage) according to the vaginal smears of OVX rats.

*Assessing OXT-mRFP1 fluorescence after oestrogen replacement.* In the fourth experiment (Exp. D, $n = 24$), we performed hormone replacement by subcutaneously implanting of oestrogen-containing tubes within the mid-back region of the rats. Female OXT-mRFP1 transgenic rats with OVX were divided into four groups ($n = 5$ 6 per group): control (sham back operation), vehicle (subcutaneous sesame oil only), low E2 replacement (20 μg β-oestradiol/ml sesame oil), and high E2 replacement (400 μg β-oestradiol/ml sesame oil)[57,58]. Low doses of E2 have previously been shown to negatively affect LH pulses but did not induce LH surges in OVX rats. High E2 levels were shown to induce an LH surge and resulted in a pro-oestrus-like state in OVX rats[57]. We confirmed the effect of low E2 and high E2 in the rats by examining their vaginal smears.

*Assessing OXT-mRFP1 gene expression after oestrogen replacement.* For the fifth experiment (Exp. E, $n = 18$), we conducted hormone replacement similarly to Exp. D. Female OXT-mRFP1 transgenic rats with OVX were divided into three groups ($n = 6$ in each group): OVX + oil only (subcutaneous sesame oil only), OVX + low E2 (low E2 replacement; 20 μg β-oestradiol/ml sesame oil), and OVX + high E2 (high E2 replacement; 400 μg β-oestradiol/ml sesame oil).

*Assessing the effects of CCK-8 administration on food intake.* For the sixth experiment (Exp. F, $n = 24$), we assessed food intake in female Wistar rats. Ovariectomised Wistar rats were divided into four groups: vehicle (subcutaneous sesame oil only) with i.p. administration of saline, vehicle (subcutaneous sesame oil only) with i.p. administration of CCK-8 (50 μg/kg body weight), high E2 with saline injection, and high E2 with CCK-8 injection. Ovariectomies were performed at 10 weeks and tube implantation was performed at 14 weeks of age; saline and CCK-8 were injected at 16 weeks. On week 16, cumulative food intake was measured at 0.5 h, 1 h, 1.5 h, 3 h, and 6 h after i.p. administration of saline and CCK-8. Prior to injections, all rats were fasted for 24 h ($n = 6$ in each sub-group). CCK-8 was administered, and the effect on food intake was assessed.

*Assessing the expression of Fos and OXT after CCK-8 administration.* For the seventh experiment (Exp. G, $n = 24$), we conducted immunohistochemistry to assess the levels of Fos and OXT in the hypothalamus of female Wistar rats. Ovariectomised rats were divided into four groups: vehicle (subcutaneous sesame oil only) with i.p. administration of saline, vehicle (subcutaneous sesame oil only) with i.p. administration of CCK-8 (50 μg/kg body weight), high E2 (400 μg β-oestradiol/ml sesame oil) with saline injection, and high E2 with CCK-8 injection. Ovariectomies were performed at 10 weeks and tube implantation was performed at 14 weeks of age; saline and CCK-8 were injected at 16 weeks. All rats were anesthetised and sacrificed at 1.5 h after i.p. administration of saline and CCK-8. Slices of brain tissues were harvested and evaluated for the expression of Fos and OXT via double-FIHC.

*Assessing the effects of administration of CCK-8 and OXTR-A on food intake.* In the eighth experiment (Exp. H, $n = 24$), we assessed food intake in female Wistar rats. Ovariectomised rats were divided into four groups: vehicle (subcutaneous sesame oil only) with i.p. administration of CCK-8 (50 μg/kg body weight) and i.c.v. administration of saline, vehicle (subcutaneous sesame oil only) with i.p. administration of CCK-8 and i.c.v. administration of OXTR-A (150 ng/μl), high E2 (400 μg β-oestradiol/ml sesame oil) with i.p. administration of CCK-8 and i.c.v.

administration of saline (sterile 0.9% saline), and high E2 with i.p. administration of CCK-8 and i.c.v. administration of OXTR-A. Ovariectomies were performed at 10 weeks, and tube implantation and i.c.v. treatments were performed at 14 weeks of age. On week 16, cumulative food intake was measured at 3 h after i.p. administration of CCK-8 and i.c.v. administration of saline or OXTR-A.

**Extraction of the hypothalamus and pituitary gland of OXT-mRFP1 transgenic rat**. Anesthetised rats were perfused transcardially with 0.1 M phosphate buffer (PB) (pH 7.4) containing heparin (1000 U/L), followed by 4% paraformaldehyde in 0.1 M PB. Rat brains and pituitaries were carefully extracted, and a small block encapsulating the hypothalamus was isolated. Blocks were post-fixed with 4% paraformaldehyde in 0.1 M PB for 48 h at 4 ℃ prepared by adding 4% paraformaldehyde to 0.1 M PB (pH 7.4) containing heparin (1000 U / L)[60]. Tissue was cryoprotected in 20% (w/v) sucrose in 0.1 M PB for 48 h at 4 ℃. Fixed tissue was cut coronally to a thickness of 30 μm using a microtome (REM-700; Yamato Kohki Industrial Co. Ltd, Saitama, Japan). The sections were divided into three groups, so that approximately the same brain region was included. The first group of sections were rinsed with 0.1 M PB and placed on glass slides. Pituitary glands were not treated.

**Evaluation of mRFP1 fluorescence in the hypothalamus and pituitary gland of OXT-mRFP1 transgenic rat**. The locations of the SON and PVN were determined according to the coordinates in the atlas of Paxinos and Watson[59]. The ap, dp, and m divisions of the PVN were divided and quantified. Sections containing the SON, PVN, and intact pituitary gland tissues were examined using a fluorescence microscope (ECLIPSE E 600; Nikon Corp., Tokyo, Japan) equipped with an mRFP1 filter (Nikon Corp.) to visualise OXT-mRFP1 expression. The images were captured with a digital camera (DS-Qi1Mc; Nikon Corp.). Using a light source of the same intensity, we averaged the mRFP1 fluorescence intensities for each region. The average mRFP1 fluorescence intensity per unit area in the SON, apPVN, dpPVN, mPVN, and PP was quantified with an imaging analysis system (NIS-Elements; Nikon Corp.).

**In situ hybridisation histochemistry**. To measure changes in the expression of the *mRFP1* and *OXT* genes in the PVN and SON via in situ hybridisation histochemistry, transgenic rats were sacrificed by decapitation after hormone replacement in the state of OVX. We performed in situ hybridisation based on previous reports[61,62]. Brain tissue was frozen immediately and carefully using crushed dry ice after decapitation. The tissues were cut with a microtome to a thickness of 12 μm to create sections for observation via in situ hybridisation. As mentioned above, the nucleus of interest was identified by referring to the rat atlas[63]. To select the sections that corresponded to those in the atlas, they were checked with dark-field microscopy. Two sections containing the SON and PVN were used from each rat to determine the autoradiography density of the autoradiographs. 35S-labelled oligodeoxynucleotide probe and protocol for in situ hybridisation histochemistry (ISH) in this experiment have been used many times before with good reliability. The details of the probe sequence and protocol for in situ hybridisation have been complementary to the transcripts encoding *OXT* and *mRFP1* (OXT probe sequence, 5′-CTC GGA GAA GGC AGA CTC AGG GTC GCA GGC-3′; mRFP1 probe sequence, 5′-GCG CGT TCG TAC TGT TCC ACG ATG GTG TAG TCC TCG TTG T-3′). The probe was 3′-end labelled by terminal deoxynucleotidyl transferase and [35S] deoxy-ATP[61,64]. The hybridised sections were exposed to autoradiography films (Amersham Hyperfilm, Buckinghamshire, UK) for 3 days (*mRFP1*) and 6 h (*OXT*), respectively. The gene expression in the obtained image was analysed semiquantitatively using Image-J software (National Institutes of Health, Baltimore, MD, USA). Using a cryostat (OTF5000, Bright Instrument Co Ltd., England), brains were sliced into 12-μm-thick coronal sections at −20 ℃. The sections were thawed after mounting them on gelatin/chrome alum-coated slides. The PVN and arcuate nucleus regions were determined by referring to the coordinates provided in the rat brain atlas.

**Fos and OXT double-FIHC**. Serial 40-μm-thick sections were rinsed twice with 0.1 M phosphate-buffered saline (PBS) and washed in 0.1 M Tris buffer (pH, 7.6) containing 0.3% Triton X-100. Sections were incubated for 72 h at 4 ℃ in primary antibody solution (goat anti-c-Fos, Santa Cruz Biotechnology, TX, USA; 1:500 or rabbit anti-OXT, Sigma-Aldrich, MO, USA; 1:5000)[65]. After washing twice in 0.3% Triton X-100 in PBS, floating sections were incubated for 24 h at 4 ℃ with a secondary antibody (Alexa Fluor 546 donkey anti-goat IgG or Alexa Fluor 488 donkey anti-rabbit IgG; Molecular Probes, OR, USA; 1:2,000 in PBS containing 0.3% Triton X-100)[65]. Sections were washed twice in PBS and then mounted on the slides and coverslipped using vectashield (Vector Laboratories Co. Ltd., CA, USA)[66]. Images of Fos+, OXT+, and Fos+/OXT+ double-labelled cells were counted manually by two researchers who were blinded to avoid bias. The number and percentage of Fos+, OXT+, and Fos+/OXT+ cells in the SON and PVN were estimated.

**Measurement of plasma OXT and oestrogen concentration**. Plasma OXT levels were measured in rat blood samples taken upon decapitation by a radioimmunoassay (RIA) method using specific anti-OXT antisera[67]. The intra- and

inter-assay coefficients of variation for measuring plasma OXT levels were 4 and 10%, respectively.

Plasma E2 levels were determined using liquid chromatography-tandem mass spectrometry (LC-MS / MS) (ASKA Pharma Medical Co. Ltd, KN, JPN)[68]. The limit of quantification of E2 was 0.025 pg/ml.

**Statistical and reproducibility**. All data points are presented as the mean ± standard error of the mean. P values were calculated by using two-tailed Student's *t* test for pair wise comparisons. An unpaired t-test was used to detect differences between male and female rats. Statistical significances were calculated based on one-way and two-way analysis of variance (ANOVA) as well as repeated-measures ANOVA, using a Tukey-Kramer-type adjustment for multiple comparisons. Correlation analyses were performed using GraphPad Prism 9. A *P*-value < 0.05 was considered statistically significant.

**Reporting summary**. Further information on research design is available in the Nature Research Reporting Summary linked to this article.

## Data availability
The source data are provided with this paper (Supplementary Data 1). Any further requests can be directed to the corresponding author.

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

## Acknowledgements

The authors would like to thank Ms. Yuki Nonaka for her animal care and technical assistance. This study was supported by Grants-in-Aid for Scientific Research (B) (No. 17H04027, Yoichi Ueta) and (C) (No. 17K08582, Takashi Maruyama) from the Japan Society for the Promotion of Science (JSPS).

## Author contributions

Study was designed by K.N. and Y.U.; The paper was written by K.N.; Experiments were performed by K.N., N.I., K.B., K.S., Y.A., H.N., K.T., S.S., H.U., M.Y. T.M., and K.Y.; Data analysis and Interpretation of data were performed by K.N., K.Y., T.H., and Y.U.; Draft and Figures are prepared by K.N.; Final Approval was made by Y.U. All authors approved the final version of the manuscript and agreed to be accountable for all aspects of the work in ensuring that questions related to the accuracy. All authors designated as authors qualify for authorship, and all those who qualify for authorship are listed.

## Competing interests

The authors declare no competing interests.
