## [Peer Review File · Communications Biology]

Reviewers' comments:

Reviewer #1 (Remarks to the Author):

Nishimura and colleagues have exploited a previously characterized transgenic rat that expresses oxytocin-monomeric red fluorescent protein 1 (Ueno et al., Physiology Reports 2020) to evaluate oestrogen-dependent oxytocin (OXT) dynamics in both male and female rats. In measuring the OXT-mRFP1 across the oestrous cycle, the authors documented that the fluorescence intensity peaked on oestrus in hypothalamic nuclei (SON and divisions of the PVN) and the posterior pituitary gland. The levels in "sham operated" males were equivalent to ovariectomized females. Not surprisingly, the authors concluded that ovariectomy decreases OXT expression. The authors did the next obvious experiment and assessed the effects of 17 β -estradiol (E2) replacement. Again not surprisingly, E2 restored OXT-mRFP1 fluorescence levels in a dose-dependent manner (high versus low dose of E2). In addition, the authors correlated E2 replacement in ovariectomized females with a reduction in food intake and body weight gain, which has been documented in mice, rats and guinea pigs. In the next experiment the authors looked at the effects of the anorexigenic hormone CCK-8 and found that it inhibited food intake and body weight gain in E2 (high dose) treated, ovariectomized females. Based on c-fos expression, there was a significant increase in the number of OXT-immunoreactive neurons expressing c-fos. Finally, the authors showed that the effects of E2 on food intake and body weight gain were blocked, in part, by an oxytocin receptor antagonist given i.c.v. In total, the studies are relatively complete and compelling, but there are a number of issues that need to be addressed:

1. The title could be more "pro-active" and describe the new discoveries of the studies.
2. The abstract should include the CCK results.
3. Is it correct that the authors used a separate cohort of females for every experiment—i.e., they never correlated the OXT-mRFP1 fluorescence with feeding? This seems like a missed opportunity.
4. It is not clear why in Experiment 6 they resorted to measuring c-fos expression in SON neurons. Did the CCK injections not enhance OXT-mRFP1 fluorescence beyond the E2 treatment? The c-fos images (Figure 5d) are not convincing. Again, it would have been more compelling to utilize the same cohort of females used for Experiment 5.
5. A one-way ANOVA for statistical analysis for Experiments 5-7 seems inappropriate since the authors are looking at group differences over time and comparing drug (CCK and OTX antagonist) effects. A two-way ANOVA would seem to be more appropriate.
6. It would help to focus the Discussion on the role of the anorexigenic hormones E2 and CCK in inhibiting food intake via the OXT signaling network. A simple circuit model would help drive home the point.

Minor points:

1. Line 237, should be "OXT neurons."
2. Line 267, should be "females compared with male rats;" and "females at prooestrous, metoestrous, dioestrous stages compared with oestrous female rats."
3. Line 463, should be "goat anti-cFos."

Reviewer #2 (Remarks to the Author):

In the present study, the authors examined the role of estrogen on oxytocin activity and function. Specifically, they assessed whether estrogen replacement in ovariectomized female rats affected oxytocin fos expression, as well as the effect of estrogen on food intake with or without CCK-8 injection. They also examined estrogen's effect on CCK-induced oxytocin activity following administration of an oxytocin-receptor antagonist. Overall, Nishimura et al. presented a thorough and multi-faceted study. Below are my suggested edits:

1. Several recent studies have also examined the interaction between estrogen and oxytocin in the control of food intake. I think a discussion of previous other literature (PMID: 31738883, 25647756, 30118729) and how your findings fit in with these studies would be beneficial.
2. The authors found that estrogen-induced oxytocin neurons in the PVH, SON and PPG. Within these areas are both magnocellular neurons and parvocellular neurons that have differing

projections. Do the authors think estrogen-induced oxytocin activity in periphery and brain are same neurons, different, or interacting?

3. While the authors showed a clear estrogen-induced increase in oxytocin neuron activity (via fos), have the authors examined binding affinity or oxytocin receptor expression?

4. I think a direct measure of an interaction between estrogen and oxytocin is necessary to conclude that oxytocin is dependent on estrogen for food intake control. The authors could try subthreshold doses of both oxytocin and estrogen. If there is a combined inhibitory effect this would demonstrate an interaction.

Reviewer #3 (Remarks to the Author):

Using OXT-mRFP1 transgenic rats, the first half of this manuscript describes changes in the fluorescence intensity of oxytocin reporter fluorescence protein (OXT-mRFP1) in the supraoptic nucleus (SON), paraventricular nucleus (PVN), and posterior pituitary during estrous cycle in intact females, as well as in ovariectomized (OVX) females and OVX females received estradiol (E2) treatment.

The second half of the manuscript describes the effects of OVX and E2 treatment on body weight and food intake in longer term, as well as the short-term effect of ip injection of CCK-8 and icv infusion of oxytocin antagonist on food intake in OVX with/without E2 treated females.

From these results, authors concluded that: 1) oxytocin expression in the hypothalamus is estrogen dependent; and 2) suppression of food intake by estrogen may be mediated by anorexigenic activity of oxytocin.

There is a number of major deficiencies and flaws in experimental design and interpretation of results.

Major concerns:

The title does not represent the content of the manuscript.

Assessment of oxytocin production in the hypothalamus and pituitary is solely by the fluorescence intensity of reporter protein. Was the fluorescence intensity correlated with the amount of oxytocin? Additional methods to validate, such as qPCR of mRNA extracted from punched tissue and/or quantitative in situ hybridization, are required.

Material and Methods does not adequately address how the fluorescence intensity was measured. It is hard for a reviewer to assess the competence of the experimental design and methods.

1. The SON in rats extends anterior to posterior direction in about 1 mm. If the brain sections were cut at 30 μ m thickness, there are ~33 sections that containing the SON. Were the measurements obtained from every section or just a few sections? The size and shape of the SON changes from anterior to posterior. More importantly for the same context, the population (density) of oxytocin neurons within the of the SON varies considerably section to section. This fact is especially important when the fluorescence intensity of brain areas rather than oxytocin neurons themselves was obtained.

2. What is the reason for measuring the fluorescence intensity of brain areas rather than oxytocin neurons? What are the limitations and reasonable interpretation of the results?

3. How were the "% of control" obtained? If a normalization method was used, it must be addressed adequately. Were the exposure time and intensity of light equal to all images obtained?

Questionable interpretation of the results.

1. Oxytocin expression in the hypothalamus is NOT estrogen dependent. In fact, OVX females still express oxytocin in the hypothalamus.

2. The notion addressed by the authors "suppression of food intake by estrogen may be mediated by anorexigenic activity of oxytocin" is probable, but extremely far-fetched considering the wide distribution of oxytocin receptor (OXTR) and estrogen receptors in the nervous system (both peripheral and central) and peripheral organs.

a) OXTR are widely distributed in the brain, including the hypothalamic area (not SON or PVN) that regulate food intake. Therefore, the icv infusion of OTA affects numbers of OXTR in the brain. Moreover, the expression of OXTR in some areas of the brain are totally estrogen dependent.

b) Estrogen receptors are widely distributed in the hypothalamus especially in the area regulating food intake. Estrogen receptors are also found in the peripheral tissues including the adipose tissue.

c) CCK receptors are widely located in the brain and peripheral organ such as the intestine. Thus, ip injection of CCK-8 would affect any of the brain areas and peripheral organs, not just oxytocin neurons.

Due to the major concerns addressed above, additional experiments and an extensive revision of the manuscript are required. For these reasons, I do not recommend the manuscript as in the current form for publication in Nature Communication.

Responses to the reviewers' comments

Manuscript No.: COMMSBIO-21-0650

Title: Oestrogen-dependent hypothalamic oxytocin expression with changes in feeding and body weight in female rats

Journal Title: *Communications biology*, Submission Date: 13th March 2021

Reviewer #1 (Remarks to the Author):

We would like to thank the reviewer for their robust and detailed assessment of our manuscript. The manuscript has been carefully revised according to the suggestions and recommendations.

The title could be more “pro-active” and describe the new discoveries of the studies.

Response: We thank the reviewer for their valuable suggestion. We have revised the title to “Oestrogen-dependent hypothalamic oxytocin expression with changes in feeding and body weight in female rats” accordingly.

The abstract should include the CCK results.

Response: We thank the reviewer for their comment. We have revised the abstract accordingly (line 33).

Is it correct that the authors used a separate cohort of females for every experiment—i.e., they never correlated the OXT-mRFP1 fluorescence with feeding? This seems like a missed opportunity.

Response: We thank the reviewer for their pertinent comment. We used different rats in each of these experiments. The body weight experiment required the use of anesthesia for micro-CT and did not use OXT-mRFP1 rats with varying fluorescence.

A one-way ANOVA for statistical analysis for Experiments 5-7 seems inappropriate since the authors are looking at group differences over time and comparing drug (CCK and OTX antagonist) effects. A two-way ANOVA would seem to be more appropriate.

Response: We thank the reviewer for their comment and apologise for this error. We have included details of a two-way ANOVA and repeated-measures ANOVA in the statistics section of the revised manuscript (Lines 399, 410, and 597).

It would help to focus the Discussion on the role of the anorexigenic hormones E2 and CCK in inhibiting food intake via the OXT signaling network. A simple circuit model would help drive home the point.

Response: We thank the reviewer for their constructive suggestion.

Minor points:

1. Line 237, should be “OXT neurons.”

Response: We thank the reviewer for their suggestion. We have revised the sentence accordingly (Line 267)

2. Line 267, should be “females compared with male rats;” and “females at prooestrous, metoestrous, dioestrous stages compared with oestrous female rats.”

Response: We thank the reviewer for their constructive suggestion. We have revised the sentence accordingly (Line 333, 334)

3. Line 463, should be “goat anti-cFos.”

Response: We thank the reviewer for their pertinent suggestions and apologise for the confusion. We have revised the manuscript accordingly (Line 585).

Reviewer #2 (Remarks to the Author):

We would like to thank the reviewer for their robust and detailed assessment of our manuscript. The manuscript has been carefully revised according to the suggestions and recommendations.

1. Several recent studies have also examined the interaction between estrogen and oxytocin in the control of food intake. I think a discussion of previous other literature (PMID: 31738883, 25647756, 30118729) and how your findings fit in with these studies would be beneficial.

Response: We thank the reviewer for their constructive suggestion. We have added one reference and revised the manuscript accordingly (Lines 280-284).

2. The authors found that estrogen-induced oxytocin neurons in the PVH, SON and PPG. Within these areas are both magnocellular neurons and parvocellular neurons that have differing projections. Do the authors think estrogen-induced oxytocin activity in periphery and brain are same neurons, different, or interacting?

Response: We thank the reviewer for their constructive comment. In this experiment, as shown in Fig. 2–4, the oestrus cycle and oestrogen affected both magnocellular neurons and parvocellular oxytocin neurons of the PVN and SON, leading to a similar result in both sets of neurons. However, it is unclear whether they are the same neurons or interact with each other.

3. While the authors showed a clear estrogen-induced increase in oxytocin neuron activity (via fos), have the authors examined binding affinity or oxytocin receptor expression?

Response: We thank the reviewer for their important comment. This study was unable to determine the binding affinity or expression of the oxytocin receptor. We believe that this is a future research topic. We have revised the discussion section to reflect this (Lines 289-290).

4. I think a direct measure of an interaction between estrogen and oxytocin is necessary to conclude that oxytocin is dependent on estrogen for food intake control. The authors could try subthreshold doses of both oxytocin and estrogen. If there is a combined inhibitory effect this would demonstrate an interaction.

Response: We thank the reviewer for their constructive suggestion. We measured the subthreshold doses of both CCK-8 (0.5 µg/ml.) and oestrogen (8 µg β-oestradiol/ml sesame oil). However, examination of the interaction at subthreshold doses did not reveal any inhibitory effect. We also performed additional experiments which also revealed no inhibitory effect.

Reviewer #3 (Remarks to the Author):

We would like to thank you for your robust and detailed assessment of our manuscript. The manuscript has been carefully revised in line with your suggestions and recommendations.

The title does not represent the content of the manuscript.

Response: We thank the reviewer for their pertinent comment. We have revised the title accordingly (“Oestrogen-dependent hypothalamic oxytocin expression with changes in feeding and body weight in female rats”).

Assessment of oxytocin production in the hypothalamus and pituitary is solely by the fluorescence intensity of reporter protein. Was the fluorescence intensity correlated with the amount of oxytocin? Additional methods to validate, such as qPCR of mRNA extracted from punched tissue and/or quantitative in situ hybridization, are required.

Response: We thank the reviewer for their constructive comment. We have added details regarding *in situ* hybridization histochemical examination with *OXT-mRFP1* mRNA in the revised manuscript. We have accordingly revised the title, introduction section, and results section.

Material and Methods does not adequately address how the fluorescence intensity was measured. It is hard for a reviewer to assess the competence of the experimental design and methods.

Response: We thank the reviewer for their constructive comment and apologise for the misunderstanding. We have deleted this sentence and revised the manuscript accordingly (Lines 518-531).

1. The SON in rats extends anterior to posterior direction in about 1 mm. If the brain sections were cut at 30 μ m thickness, there are ~33 sections that containing the SON. Were the measurements obtained from every section or just a few sections? The size and shape of the SON changes from anterior to posterior. More importantly for the same context, the population (density) of oxytocin neurons within the of the SON varies considerably section to section. This fact is especially important when the fluorescence intensity of brain areas rather than oxytocin neurons themselves was obtained.

Response: We thank the reviewer for their pertinent comments and apologise for the confusion. The tissue was cut into 30- μ m-thick sections with a microtome for observation. The sections were divided into three groups, so that approximately the same brain region was included. Using the first group of sections, mRFP1 fluorescence in the hypothalamus was observed and photographed using a fluorescence microscope. We have revised the manuscript accordingly (Lines 548-550).

2. What is the reason for measuring the fluorescence intensity of brain areas rather than oxytocin neurons? What are the limitations and reasonable interpretation of the results?

Response: We thank the reviewer for their pertinent questions. In this study, we quantitatively evaluated OXT expression in OXT-mRFP1 transgenic rats by measuring fluorescence intensity as reported previously (Arase et al., 2018; Katoh et al., 2011). Therefore, the fluorescence intensity was measured and quantitatively evaluated.

3. How were the "% of control" obtained? If a normalization method was used, it must be addressed adequately. Were the exposure time and intensity of light equal to all images obtained?

Response: We thank the reviewer for their important comment. Using a light source of the same intensity, we averaged the mRFP1 fluorescence intensities for each region. We have clarified this in lines (Lines 559-560).

Questionable interpretation of the results.

1. Oxytocin expression in the hypothalamus is NOT estrogen dependent. In fact, OVX females still express oxytocin in the hypothalamus.

Response: We thank the reviewer for their comment. Oestrogen regulates changes in OXT expression in hypothalamic OXT neurons in a dose-dependent manner. To avoid confusion, we have changed the wording to reflect this in line (Line 32 and 132).

2. The notion addressed by the authors "suppression of food intake by estrogen may be mediated by anorexigenic activity of oxytocin" is probable, but extremely far-fetched considering the wide distribution of oxytocin receptor (OXTR) and estrogen receptors in the nervous system (both peripheral and central) and peripheral organs.

a) OXTR are widely distributed in the brain, including the hypothalamic area (not SON or PVN) that regulate food intake. Therefore, the icv infusion of OTA affects numbers of OXTR in the brain. Moreover, the expression of OXTR in some areas of the brain are totally estrogen dependent.

Response: We thank the reviewer for their important comments. This study did not examine oxytocin or oestrogen receptors. It is unknown whether OXTR expression is oestrogen-dependent. We believe that this is worthy of investigation in future studies.

b) Estrogen receptors are widely distributed in the hypothalamus especially in the area regulating food intake. Estrogen receptors are also found in the peripheral tissues including the adipose tissue.

Response: We thank the reviewer for their helpful comments. Oestrogen receptors are also found in peripheral tissues, including adipose tissue. Therefore, we believe that there is a change in body weight. However, this study did not examine the oestrogen receptor. It is stated in the limitation (Lines 289-290).

c) CCK receptors are widely located in the brain and peripheral organ such as the intestine. Thus, ip injection of CCK-8 would affect any of the brain areas and peripheral organs, not just oxytocin neurons.

Response: We thank the reviewer for their constructive suggestion. As the reviewer has correctly mentioned, CCK receptors are widely present in peripheral organs such as the brain and intestine. Therefore, various effects are expected. However, in this experiment, the relevant examinations were performed under the same conditions whereby CCK-8 was administered.

Reviewers' comments:

Reviewer #1 (Remarks to the Author):

The authors, Nishimura and colleagues, have addressed most of my previous criticisms: the title is more compelling, they have incorporated the CCK results into the abstract, albeit one sentence, and the appropriate statistical analyses has been used (with a minor exception noted below). However, the Discussion could still be more focused on the literature that is more pertinent to the current results. For example, lines 207-214 discusses the role of oxytocin in pain modulation during parturition, and lines 229-236 discusses the loss of oestrogen in menopause (surgical) and the potential effects on OTX production. What is more relevant to the present findings and of greater interest to the readers is a discussion of the effects of oxytocin on hypothalamic feeding circuits—specifically the effects of OTX on the anorexigenic POMC and the orexigenic NPY neurons. Hence, my original comment, “It would help to focus the Discussion on the role of the anorexigenic hormones E2 and CCK in inhibiting food intake via the OXT signaling network,” has not been addressed. Also, it would be of interest to the readers if the authors could elaborate on how oestrogen up-regulates OTX expression. Is it via a classical (i.e., ERE dependent) signaling pathway in oxytocin neurons?

1. Panel i (bar graphs) is missing in Figure 1.

Minor points:

1. Line 90: use the past tense, the difference in body weight was.....
2. Line 92, should read: We observed a significant change in feeding that was dependent...
3. Line 223: oestrus should be pro-oestrus.
4. Line 280 should read: involved in the change of fat.....
5. Lines 280-284: the last concluding sentences do not make any sense
6. Line 307: the authors should use a t-test rather than a one-way ANOVA.
7. Line 481: ration should be ratio.
8. Line 505: oestrus should be pro-oestrus.
9. Figure 6: the green line representing the OVX+High E2+Saline group is missing.

Reviewer #3 (Remarks to the Author):

In this revised manuscript, two figures were added from additional studies requested by reviewers. However, the manuscript was not fundamentally changed. In fact, none of concerns raised by this reviewer was adequately addressed in the revised manuscript. Therefore, the exact same concerns remained.

The photomicrographs provided in the Figure 5 show nothing in convincing manner. Therefore, the numerical data provided in the figure are also very questionable.

Responses to the reviewers' comments

Manuscript No.: COMMSBIO-21-0650A

Title: Oestrogen-dependent hypothalamic oxytocin expression with changes in feeding and body weight in female rats

Journal Title: *Communications biology*, Submission Date: 23th November 2021

Reviewer #1 (Remarks to the Author):

We would like to thank the reviewer for their robust and detailed assessment of our manuscript. The manuscript has been carefully revised according to the suggestions and recommendations.

“It would help to focus the Discussion on the role of the anorexigenic hormones E2 and CCK in inhibiting food intake via the OXT signaling network,” has not been addressed. Also, it would be of interest to the readers if the authors could elaborate on how oestrogen up-regulates OTX expression. Is it via a classical (i.e., ERE dependent) signaling pathway in oxytocin neurons?

Response: We thank the reviewer for their comment and apologise for the non-answers. “Oestrogen, OXT and CCK-8 decreases food intake. CCK-8 acts on OXT neurons in the hypothalamus to decreases food feeding. In this experiment, the addition of oestrogen to CCK-8 resulted in more decreased food intake and more activation of OXT neurons. Oestrogen proves that OXT neurons are more activated. And in this study, oestrogen was found to regulate OXT neurons in transgenic rats via in situ experiments. The oestrogen receptor ER β is localised in the PVN and SON of the hypothalamus¹². Therefore, oestrogen may act on PVN and SON in the hypothalamus via ER β to activate OXT neurons.” We have revised the discussion section to reflect this (Lines 315-320).

Panel i (bar graphs) is missing in Figure 1.

Response: We thank the reviewer for their comment and apologize for the mistake. We corrected the figure 1.

Line 90: use the past tense, the difference in body weight was.....

Response: We thank the reviewers for their constructive comments. We corrected the sentence (line 90).

Line 92, should read: We observed a significant change in feeding that was dependent...

Response: We thank the reviewer for their comment and apologise for this mistake. We corrected the sentence (line 93).

Line 223: oestrus should be pro-oestrus.

Response: We thank the reviewer for their comment and apologise for this mistake. We corrected the sentence (line 254).

Line 280 should read: involved in the change of fat.....

Response: We thank the reviewer for their suggestion. We have revised the sentence to “In this experiment, oestrogen was associated with fat composition. However, it is unclear whether OXT was directly involved.” (line 312-313)

Lines 280-284: the last concluding sentences do not make any sense

Response: We thank the reviewer for their important comment. As you say, this sentence “In this study, oestrogen was found to regulate OXT neurons in transgenic rats via *in situ* experiments. However, the relationship between direct feeding of OXT remains unknown.” has been deleted.

Line 307: the authors should use a t-test rather than a one-way ANOVA.

Response: We thank the reviewer for their suggestion. We have revised the sentence accordingly (line 342 and 670-671)

Line 481: ration should be ratio.

Response: We thank the reviewer for their suggestion. We have revised the word accordingly (line 542)

Line 505: oestrus should be pro-oestrus.

Response: We thank the reviewer for their important comments. We corrected the sentence (line 565).

Figure 6: the green line representing the OVX+High E2+Saline group is missing.

Response: We thank the reviewer for their comment and apologise for this error. We corrected the figure 6.

Reviewer #3 (Remarks to the Author):

We would like to thank the reviewer for their robust and detailed assessment of our manuscript. The manuscript has been carefully revised according to the suggestions and recommendations.

In this revised manuscript, two figures were added from additional studies requested by

reviewers. However, the manuscript was not fundamentally changed. In fact, none of concerns raised by this reviewer was adequately addressed in the revised manuscript. Therefore, the exact same concerns remained.

The photomicrographs provided in the Figure 5 show nothing in convincing manner. Therefore, the numerical data provided in the figure are also very questionable.

Response: We thank the reviewer for their helpful comments. We made a correlation diagram between fluorescence, mRNA, and plasma to obtain more reliable results of figure. There was a correlation between E2 and OXT fluorescence. There was also a significant correlation between the OXT mRFP1 gene and plasma E2 and OXT. (Figure 4-6)

Reviewers' comments:

Reviewer #3 (Remarks to the Author):

The major claims of this paper are not well described. Are they the effect of OVX and E2 therapy on body weight and food intake, estrogen depend expression of oxytocin, intensity of mRFP1 as an indicator of oxytocin expression, the effect of CCK-8 on food intake and FOS expression in oxytocin cells, or ICV infusion of OXTR-A on body weight and food intake? Each finding is fine independently; however, these findings were not connected well due to the lack of target specific approaches. For examples, E2 therapy, CCK-8 injection, and ICV infusion of OXTR-A affect many brain areas not only regulating food intake but also many other functions. Therefore, it is extremely hard to make a case. Indeed, it appears that the authors did not make a strong case in this manuscript. That may be why the title of this manuscript does not accurately describe the contents of this study. Most of findings such as the effect of E2, OVX, CCK-8, and OXTR-A on food intakes were not novel.

Responses to the reviewers' comments

Manuscript No.: COMMSBIO-21-0650B

Title: Oestrogen-dependent hypothalamic oxytocin expression with changes in feeding and body weight in female rats

Journal Title: *Communications biology*, Submission Date: 28th June 2022

Reviewer #3 (Remarks to the Author):

We would like to thank the reviewer for their robust and detailed assessment of our manuscript. The manuscript has been carefully revised according to the suggestions and recommendations.

The major claims of this paper are not well described. Are they the effect of OVX and E2 therapy on body weight and food intake, estrogen depend expression of oxytocin, intensity of mRFP1 as an indicator of oxytocin expression, the effect of CCK-8 on food intake and FOS expression in oxytocin cells, or ICV infusion of OXTR-A on body weight and food intake? Each finding is fine independently; however, these findings were not connected well due to the luck of target specific approaches. For examples, E2 therapy, CCK-8 injection, and ICV infusion of OXTR-A affect many brain areas not only regulating food intake but also many other functions. Therefore, it is extremely hard to make a case.

Response: We thank the reviewer for their suggestions. As noted, E2 therapy, CCK-8 peripheral injections, and ICV injections of OXTR-A affect many brain regions that regulate not only food intake and various physiological functions. However, the difference in food intake related to peripheral or central administrations of E2, CCK8, and OXTR-A was investigated in ovariectomized female rats. This study considers “differences” and “changes” among each experiment. We have included "Change" in the title because we focus on the changes in the observed phenomenon. “The advantage of this study was the quantitative demonstration of hypothalamic-pituitary OXT system by examining OXT blood concentrations, correlation coefficient, and OXT mRNA expression levels in the SON and PVN. We confirmed that feeding suppression induced by the peripheral administration of CCK-8 resulted in a difference in the activation of OXT neurons, which was enhanced among oestrogen-replaced OVX female rats. Therefore, we investigated “changes” in food intake by combining CCK-8, oestrogen, and OXTR-A administration and proved the relationship between oestrogen and OXT.” We have revised the discussion section to reflect this (Lines 220-226).

Indeed, it appears that the authors did not make a strong case in this manuscript. That may be

why the title of this manuscript does not accurately describe the contents of this study. Most of findings such as the effect of E2, OVX, CCK-8, and OXTR-A on food intakes were not novel.

Response: We thank the reviewers for their constructive comments. As pointed out, most findings, such as the effects of E2, OVX, CCK-8, and OXTR-A on food intakes in female rats, might not be novel. However, the difference in food intake of E2, CCK8, and OXTR-A was investigated in our experiment in OVX female transgenic rats that expressed the OXT-mRFP1 fusion gene, as mentioned above. There were no studies on the combination of E2, CCK8, and OXTR-A. The novelty of this study was the success in identifying oestrogen-dependent hypothalamic-pituitary OXT changes by observing the fluorescent intensities of mRFP1 in OXT neurons and their axon terminals in the PP. To prove the change in fluorescence, we examined E2 and OXT blood concentrations, correlation coefficient and OXT mRNA expression level and quantitatively proved hypothalamic-pituitary OXT fluorescent intensities changes. We have revised the sentence accordingly. (Lines 213-215, 223 and 322-323)